# Impact of modified turbulent diffusion of PM$_{2.5}$ aerosol in WRF-Chem simulations in Eastern China

Wenxing Jia[1,2], Xiaoye Zhang[2,3*]

[1]Key Laboratory for Aerosol-Cloud-Precipitation of China Meteorological Administration, Nanjing University of Information Science & Technology, Nanjing, 210044, China

[2]Key Laboratory of Atmospheric Chemistry of CMA, Chinese Academy of Meteorological Sciences, Beijing, 100081, China

[3]Center for Excellence in Regional Atmospheric Environment, IUE, Chinese Academy of Sciences, Xiamen, 361021, China

Correspondence to: X. Zhang (xiaoye@cma.gov.cn)

## Abstract

Correct description of the boundary layer mixing process of particle is an important prerequisite to understanding the mechanism of heavy pollution episodes. Turbulent mixing process of particles is usually denoted by the turbulent diffusion relationship of heat, meaning that the turbulent transport of particles and heat are similar. This similarity has, however, never been verified. Here we investigate the dissimilarity between particles and heat, indicating that the unified treatment of all scalars in the model is questionable. Using mixing-length theory, the turbulent diffusion relationship of particle is established, embedded in the model and verified on a long-term scale. Simulated results of PM$_{2.5}$ concentration were improved by 8.3% (2013), 17% (2014), 11% (2015) and 11.7% (2017) in Eastern China, respectively. However, under the influence of complex topography, the turbulent diffusion process is insensitive to the simulation of the pollutant concentration. In addition to the PM$_{2.5}$ concentration, the simulation of the CO concentration has also been improved, which shows that the turbulent diffusion process is extremely critical to the change in the concentration of pollutants.





# 1 Introduction

Along with the intensive urbanization and tremendous economic development, numerous incidents of aerosol pollution have frequently occurred in China (An et al., 2019; Q. Zhang et al., 2019; X. Zhang et al., 2019). Aerosol pollution, characterized by $PM_{2.5}$, occurs primarily in the planetary boundary layer (PBL). The horizontal transportation and vertical distribution of pollutants are obviously affected by the PBL mixing process, associated with intricate turbulence eddies (Liu et al., 2018; Ren et al., 2018; Wang

et al., 2018; Du et al., 2020). Turbulent transport, as a vital process, controls the exchange of momentum, heat, water vapor and pollutants through turbulence eddies within the PBL (Stull, 1988).

Turbulent transports of temperature, water vapor and $CO_2$ has long been considered similar (Kays et al., 2005). However, this statement is usually invalid and is regarded as applicable only under neutral stratifications. Previous researchers have demonstrated that temperature-humidity dissimilarity, and such

a disparity between the effectiveness of heat and water vapor transport, is due to different mechanisms of scalar transport (Katul et al., 2008; van de Boer et al., 2014; Guo et al., 2020).For example, the effect of advection (Assouline et al., 2008), entrainment at the top of PBL (Cava et al., 2008; Gao et al., 2018) and heterogeneity in sources and sinks (Detto et al., 2008; Wang et al., 2014; Guo et al., 2016). Moriwaki and Kanda (2006) also indicated that the differences of turbulent transport between heat and $CO_2$ were

due to both by the active role of temperature and the heterogeneity of the source distribution. Li and Bou-Zeid (2011) revealed that the transport dissimilarity between the momentum and the scalar likely resulted from the topology of turbulent structures. As a result, there are differences between turbulent transport of vectors and scalars, or between scalars. However, less attention has been paid to turbulent transport of particles. Dupont et al. (2019) have proven that the turbulent dissimilarity of transport between dust, heat

and momentum. The only studies assumed that particles were considered passive scalars with the same source/sink as heat, and that they used similarity to correct particle flux from the heat flux (Damay et al., 2009; Deventer, Held et al., 2015). A key question is whether the turbulent transport between temperature and particles is similar. This similarity has, however, never been verified, due to the lack of observational turbulence data of particles.



The turbulent diffusion processes of all scalars (including active and passive scalars) are dealt with in a unified manner in the current model. To date, only a few studies have shown that pointed out the meteorological fields and pollutants can be changed by adjusting the minimum value of turbulent diffusion coefficient (TDC) (Savijarvi et al., 2002; Wang et al., 2018; Du et al., 2020; Liu et al., 2021), increasing turbulent kinetic energy (TKE) (Foreman and Emeis, 2012) and modifying experiment

expressions (Sušelj and Sood, 2010; Huang and Peng, 2017). Recently, Jia et al. (2021) obtained the TDC of particles by using high-resolution vertical flux data of particles based on the mixing length theory. Additionally, this relationship has been embedded into the WRF-Chem model to calculate the PBL mixing process of pollutants separately. This work has initially improved the overestimation of pollutant concentration at night in winter 2016 in Eastern China. However, a series of heavy pollution incidents

have occurred and attracted much attention since 2013 (Yang et al., 2018; Zhong et al., 2019). Therefore, we conducted a series of simulations for the heavy pollution periods in winter from 2013 to 2017 in this study. The difference between this study and previous work is that previous work focused on the analysis of observations, while this study mainly explores the uncertainty of the influence of the model on the turbulent diffusion of particles.

## 2 Data and methods

### 2.1 Data

In this study, the aerosol pollution level is denoted by the hourly surface $PM_{2.5}$ concentration that is available from the official website of the China National Environmental Monitoring Center from 1 January 2013 to 31 January 2017. $PM_{2.5}$ concentration stations increased from 35 cities in

2013(illustrated by red dots in Fig. 1b) to 78 cities in 2017(illustrated by black dots in Fig. 1b) in Eastern China. In addition to $PM_{2.5}$ observations, the hourly concentrations of CO were acquired from the National Air Quality real-time publication platform (http://106.37.208.233:20035, last access: 20 May 2021). Aside from this, the hourly meteorological observation data, including temperature, pressure, relative humidity, wind and visibility from the national automatic weather stations (AWS) provided by

the National Meteorological Information Center of China Meteorological Administration (NMICMA) (illustrated by gray crosses in Fig. 1b). The time period of the data selected is from 1 January 2013 to 31





January 2017. In addition, observational turbulence data are obtained from the Pingyuan County

Meteorological Bureau (37.15°N, 116.47°E), Shandong Province, from 27 December 2018 to 8 January

2019 (illustrated by orange triangle in Fig. 1b). Identical eddy-covariance systems were operated,

including three-dimensional sonic anemometer-thermometer (IRGASON, Campbell Scientific, USA)

and $CO_2/H_2O$ open-path gas analyzer (LI7500, LI-COR, USA). These instruments measured three

components of wind speed, potential temperature, water vapor and $CO_2$ concentrations with a frequency

of 10 Hz. The turbulence data finally was split into 30-min segments. In addition, a continuous particle

measuring instrument E-sampler () and a high-frequency sampling visibility sensor CS120A () were used

to obtain PM2.5 mass concentration every minute and visibility of 1 Hz. The calculation of 30-min

vertical flux of $PM_{2.5}$ is based on the nonlinear relationship between $PM_{2.5}$ concentration and visibility

(Ren et al., 2020). Detailed background and calculation principle of this method were presented in Ren

et al. (2020), so we only describe key steps. Firstly, we separate $PM_{2.5}$ concentration (C) and visibility

datasets (V) into mean and turbulent deviations (i.e., $c = \overline{c} + c'$ and $V = \overline{V} + V'$). Secondly, we get

the fitted coefficients by using exponential correlation between the $PM_{2.5}$ concentration and visibility

(i.e., $c = a \cdot V^b$). Thirdly, combining the first two steps, we can get the turbulent fluctuations of $PM_{2.5}$

concentration (i.e., $c' = a \cdot (\overline{V} + V')^b - \overline{c}$). Finally, we use fluctuations of vertical velocity (i.e., $w'$)

and of PM2.5 concentration (i.e., $c'$) to calculate the vertical flux of $PM_{2.5}$ (i.e., $\overline{w'c'}$).

To investigate the influence of the PBL height (PBLH) on the $PM_{2.5}$ pollution, soundings collected at the

Fuyang site (32.54°N,115.5°E) and the Anqing site (30.37°N,116.58°E) (illustrated by yellow pluses in

Fig. 1b) for the period 2013-2017 were analyzed. These two stations are equipped with L-band

radiosonde systems (Miao et al., 2018), which proved a fine resolution (1 Hz) profiles of temperature,

relative humidity and wind speed two times (0800 and 2000 BJT) a day during winter. To eliminate the

error caused by the difference of calculation methods of PBLH, Richardson number method is used to

calculate the PBLH in both observation and simulation. The Richardson number is defined as follows:

$$Ri(z) = \frac{g\left(\theta_{vz} - \theta_{vs}\right)\left(z - z_s\right)}{\theta_{vs}\left(u_z - u_s\right)^2 + \left(v_z - v_s\right)^2} \tag{1}$$



where $z$ is the height above ground, g is the gravity, $\theta_v$ is the virtual potential temperature, and $u$ and $v$ are the component of wind. The subscript "s" denotes the surface level. The height at which the Richardson number equals 0.25 is defined as the PBLH.

## 2.2 Numerical simulation

Long-term three-dimensional simulation experiments are enforced in this study from the winter of 2013 to 2017, when Eastern China frequently experienced severe and persistent aerosol pollution events (Zhong et al., 2019). One month for each winter from 2013 to 2017 was selected, and a total of four months were confirmed, which are January 2013, December 2014, December 2015 and January 2017, respectively. The anthropogenic emissions of BC, OC, CO, $NH_3$, $NO_x$, $PM_{2.5}$, $PM_{10}$ and volatile organic compounds (VOCs) are set based on the monthly Multi-resolution Emission Inventory for China (MEIC) from 2013 to 2017 are provided by Tsinghua University, with a resolution of 0.25°×0.25° (http://meicmodel.org/, last access: 20 May 2021). The model domain was centered over Eastern China with a horizontal resolution of 33 and 6.6 km (Fig. 1a). The model top was set to the 50 hPa level, and 48 vertical layers were configured below the top. To resolve the PBL structure, 21 layers below 2 km (AGL) were set. The physics parameterization schemes selected for this study included the Morrison double-moment microphysics scheme (Morrison et al., 2009), RRTMG longwave/shortwave radiation schemes (Iacono et al., 2008), MM5 similarity surface layer scheme (Jiménez et al., 2012), Noah land surface scheme (Chen and Dudhia, 2001), Singer-layer UCM scheme (Kusaka et al., 2001), CLM4.5 lake physics scheme (Subin et al., 2012; Gu et al., 2015), ACM2 planetary boundary layer scheme (Pleim, 2007), Grell-3D cumulus scheme (Grell and Devenyi, 2002). And the chemical mechanism is the RADM2-MADE/SORGM scheme (Ackermann et al., 1998; Schell et al., 2001). The initial and boundary conditions of meteorological fields were set up using the National Centers for Environmental Prediction (NCEP) global final (FNL) reanalysis data, with a resolution of 1 ° × 1 ° (https://rda.ucar.edu/datasets/ds083.2/, last access: 20 May 2021). And the initial and boundary conditions of chemical fields were configured using the global model output of Model for Ozone and





Related Chemical Tracers (MOZART) (http://www.acom.ucar.edu/wrf-chem/mozart.shtml, last access: 20 May 2021).

Simulation using abovementioned configurations is referred to as the original runs. In the original PBL parameterization scheme, TDCs of heat and momentum are different (i.e., $K_h \neq K_m$), the turbulent mixing process of pollutants is similar to that of heat, which supposes the eddy diffusions of particles and heat are identical (i.e., $K_h = K_c$). While in the improved scheme, the eddy diffusion of particles is calculated by the TDC of particles (i.e., $K_c$), which is different from that of heat (i.e., $K_c \neq K_h$). These improved experiments are regarded as the new runs hereafter. All simulation included a total of eight months. The

91-h simulation is conducted beginning from 0000UTC of three days ago for each day (i.e., 248 simulation experiments), and first 64-h of each simulation is considered as the spin-up period.

The TDC is parameterized by the mixing length ($l$) and the function of Richardson number ($f(Ri)$), that is

$$K = 0.01 + \sqrt{ss} \cdot l^2 \cdot f(Ri) \qquad (2)$$

where $ss$ is the wind shear (i.e., $ss = (\partial u/\partial z)^2 + (\partial v/\partial z)^2$), 0.01 refers to the minimum value of TDC

in the model, and the mixing length formula (i.e., $l = \kappa z/(1 + \kappa z/\lambda)$, $\lambda$=80) proposed by Blackadar (1962) is widely used in the model (Louis, 1979; Liu and Carroll, 1996; Lin et al., 2008; Pleim, 2016). Many previous studies have showed various functions of Richardson number, which represent the different situations of turbulence.

(i) For the stable conditions (i.e., Ri≥0), Esau and Byrkjedal (2007) suggested:

$$f_h = (1 + 10Ri + 50Ri^2 + 5000Ri^4)^{-1} + 0.0012 \qquad (3)$$

$$f_m = 0.8 f_h + 0.00104 \qquad (4)$$

where $f_h$ and $f_m$ denote the function of heat and momentum, respectively, and these functions have been implemented in the model. We added an additional function of particles into the model, that is





$$f_c = \left(1 + 66.6 Ri\right)^{-1} \tag{5}$$

which is used to denote the turbulent mixing process of particles within the PBL. For detailed analysis and comparison of functions, please refer to Jia et al. (2021).

(ii) For the unstable conditions (Ri<0), Equation (2) is rewritten as:

$$K_h = 0.01 + \sqrt{ss \cdot (1 - 25Ri)} \cdot l^2 \tag{6}$$

$$K_m = 0.8 \cdot K_h \tag{7}$$

Considering that the pollution is usually accompanied by the stable boundary layer, we mainly modify the program of the stable boundary layer, while for the unstable boundary layer, we still use the default program of the original scheme.

There are several important things to note about the TDC of particles. (1) It is calculated by the explicit local gradient to represent the PBL mixing process of particles, which are more suitable in the stable

boundary layer (SBL). (2) The new scheme avoids the inapplicability of the Monin-Obukhov similarity theory (MOST), the deviation of the PBLH in the SBL, higher computational efficiency (Li et al., 2010), and it is easier to apply to forecasting models in the future. (3) It is used to evaluate the PBL mixing process of pollutants separately, which can improve the simulation results of pollutants and does not deteriorate the simulation results of meteorological parameters.

## 3 Temperature-particles transport dissimilarity

The PM$_{2.5}$ concentration frequently reached hazardous levels above 100 μg m$^{-3}$ during six heavy pollution episodes (marked by HPE1-HPE6 in Fig. S1). The turbulent characteristics of PM$_{2.5}$ concentration have been demonstrated in Ren et al. (2020), and the turbulence characteristics of heat and particles were markedly different (Jia et al., 2021; Ren et al., 2021). Based on the previous studies, the turbulent

correlation coefficient is used to evaluate the transport efficiencies for heat, water vapor, momentum and particles (Stull, 1988; Li and Bou-Zeid, 2011; Dupont and Patton, 2012). The expression as follows:



$$R_{wp} = \frac{\overline{w'p'}}{\sigma_w \sigma_p} \qquad (8)$$

$R_{wp}$ denotes the correlation coefficient between the fluctuations of $w'$ and $p'$, while $p$ stands for the temperature $T$, specific humidity $q$, longitudinal velocity component $u$ and particles $c$. This value is between -1 to 1 (negative correlation to positive correlation), and zero indicates that the two parameters

are uncorrelated. The $\sigma_w$ and $\sigma_p$ are the standard deviations of vertical velocity and parameter $p$ (i.e., $T$, $q$, $u$, $c$) over a 30-min interval, respectively. If the MOST is applicable, it indicates the turbulent mechanisms of heat, water vapor and particles are the same, i.e., $R_{wt}=R_{wq}=R_{wc}$ (Liu et al., 2017). Previous studies have investigated different mechanisms of scalar transport between temperature and humidity (Moriwaki and Kanda 2006; Katul et al., 2008; van de Boer et al., 2014; Guo et al., 2016, 2020). Lacking

profile data for the PM$_{2.5}$ concentration (Yuan et al., 2019; Ren et al., 2020), there is little about the transport efficiency of fine particles (i.e., PM$_{2.5}$). The correlation coefficient of heat flux and particle flux can be defined as:

$$R_{wt,wc} = \frac{\overline{\left(w't' - \overline{w't'}\right)\left(w'c' - \overline{w'c'}\right)}}{\sigma_{w't'}\sigma_{w'c'}} \qquad (9)$$

$\sigma_{w't'}$ and $\sigma_{w'c'}$ are the standard deviations of $w't'$ and $w'c'$, respectively. The correlation coefficients of the heat ($R_{wt}$), fine particles ($R_{wc}$), and heat flux and particle flux ($R_{wt,wc}$) are presented in Fig. 2. Clearly,

there is an obvious difference between $R_{wt}$ and $R_{wc}$. Whether transport efficiency is $R_{wt}$ or $R_{wc}$, transport efficiency can exhibit the greatest variability at night during the HPEs, probably suggesting an increasing complexity of turbulent structures at night (Fig. 2b and 2c). High correlation exists between heat and fine particles fluxes at night (especially at the wee hours) in the HPEs (Fig. 2d), which indicates that these fluxes are performed by the same motions within the PBL. Previous research has noted that the

atmospheric vertical mixing is mainly controlled by the large-scale eddies' percentage at night during the HPEs (Li et al., 2020). However, it should be mentioned that the correlation coefficient between heat and fine particles fluxes ($R_{wt,wc}$) changes dramatically at night (Fig. 2d). This means that these two fluxes transported with different eddies in a short time, or transported at different time periods by the same eddy when the correlations diminish. Consequently, there is a difference between the transport of heat and fine





particles fluxes. Whether scalar is temperature or particle, it is debatable that the mixing process of all

scalars are dealt with a unified manner within the PBL. As a result, we urgently need to develop a TDC

of particles, which is used only to calculate the mixing process of pollutants within the PBL.

## 4 Improvement of PM$_{2.5}$ concentration

Based on the TDC relationship of particles in the previous study (Jia et al., 2021), this study applies this

relationship to a long-term scale simulation for verification. Figure 3 shows the average value of

simulated and observed PM$_{2.5}$ concentration at night from 2013 to 2017, and the PM$_{2.5}$ concentration was

overestimated to varying degrees in Eastern China. The relative bias (RB) in the mean regional value is

as high as 11.8% (2013), 48% (2014), 23.8% (2015) and 20.9% (2017), respectively (Fig. 3i-l).

Compared to the original scheme, the new scheme improves the situation where the pollutant

concentration is overestimated at night in Eastern China (Fig. 4a-d). The degree of overestimation of the

pollutant concentration is reduced, and the relative bias of average value of the new scheme is 3.5%

(2013), 31% (2014), 12.8% (2015) and 9.2% (2017), respectively (Fig. 4e-h). Moreover, the absolute

bias in the mean value is reduced by 8.3% (2013), 17% (2014), 11% (2015) and 11.7% (2017),

respectively (Fig. 4i-l). To better evaluate the model performance, figure 5 shows the Taylor diagram of

hourly PM$_{2.5}$ concentration, and the black (red) dots indicate original (new) simulation results at all

stations from 2013 to 2017. The statistical results present a consistent feature, that is, the worse the

simulation results of the original scheme are, the more obvious improvement of the new scheme becomes

(arrows indicate improved stations in the Fig. 5). The results indicate that the pollutant concentrations at

all stations are not improved to the same extent. When the original scheme overestimates the pollutant

concentrations, the new scheme will reduce the degree of overestimation. While the pollutant

concentrations are underestimated by the original scheme, the new scheme does not increase the degree

of underestimation again (Fig. 5). And the standard deviation (normalized) of the mean value is decreased

by 0.2 (2013), 0.28 (2014), 0.14 (2015) and 0.16 (2017) (Fig. 5). As a whole, the new scheme can

significantly improve the common phenomenon of overestimated pollutant concentration in the SBL in

Eastern China (Fig. 5).



In addition to the changes in the pollutant concentration near the surface, we should also pay attention to the changes in the pollutant concentration in the vertical direction. Theoretically, increasing turbulent diffusion will reduce the pollutant concentrations near the surface-layer, and the pollutants will be more fully mixing in the vertical direction, which results in lower concentrations of pollutants in the near surface-layer and higher concentrations of pollutants in the upper layer. Actually, the pollutant concentration is reduced in the surface-layer and it is increased in the upper layer at night (Fig. 6), which is consistent with the theory.

## 5 Uncertainty analysis

### 5.1 Meteorological parameters

Depending on the transport dissimilarity of heat and particles, the TDC of particles was added separately in the model to calculate the turbulent mixing process of particles. For correctional approaches, it is important that a new scheme does not lead to worse performance than that with the original scheme. To verify the new scheme without affecting the simulation results of the meteorological parameters, the simulation results of the near-surface meteorological elements (i.e., 2-m temperature, 2-m relative humidity and 10-m wind speed) have been compared and analyzed. It can be seen from Figure S2-S4 that the correlation coefficients of meteorological parameters by two schemes are greater than 0.99, noting that the new scheme does not alter the performance of meteorological fields, which is an advantage of the new scheme. Compared with previous studies, modifying the turbulent diffusion coefficient of heat not only affects the simulation of temperature (Savijarvi and Kauhanen, 2002), but also influences the results of pollutants (Liu et al., 2021). Improving the parameterization scheme is a long and tough process, making it difficult to improve the simulation results of all parameters at once. When the simulation results of one parameter are improved, we should first seek to ensure that the simulation results of other parameters are not deteriorated. Then, we are going to look at improving other parameters. Although the aerosol-radiation two-way feedback process has been considered in the atmospheric-chemistry two-way coupled model, the mean fractional change in $PM_{2.5}$ concentration varying just a few percent (Li et al., 2017; Wu et al., 2019; Gao et al., 2020). We should focus more on the feedback process between turbulence and aerosol, and hopefully develop a turbulence-aerosol two-way feedback module.



Some turbulent characteristics (e.g., turbulence barrier effect) can be taken into consideration during the HPEs, reflecting a more realistic pollutant concentration evolution process. We think the next step is to

solve this major problem.

## 5.2 PBL height

Although PBL height (PBLH) is widely used to determine the effective air volume and atmospheric environmental capacity for pollutant diffusion (Miao et al., 2018), the influence of PBLH on the pollution is uncertain. (1) There are various methods to determine the PBLH, either through observation or

simulation (Jia and Zhang., 2020; Zhang et al., 2020). Various methods diagnose different PBLH, which reinforces uncertainty about the PBLH as a criterion. (2) There does not necessarily reflect a negative correlation between pollutant concentration and PBLH. The relationship between the PBLH and $PM_{2.5}$ pollution has been revealed on the basis of the four-year radiosonde measurements, and the results show that the correlation between PBLH and $PM_{2.5}$ concentration is different in various regions (Miao et al.,

2018). Moreover, when the PBLH is higher, the corresponding pollutant concentration is not necessarily lower (Miao et al., 2021). When there is a transport stage during the HPEs with a high wind speed, the mechanical turbulence is strong, and the PBLH and pollutant concentration increase simultaneously. Therefore, the relationship between PBLH and $PM_{2.5}$ pollution is intricate. The impact of PBLH is ultimately represented through the TDC in the model, because the PBLH is used to calculate TDC. If the

pollutant concentration is clearly controlled by the PBLH, when the pollutant concentration is overestimated and the PBLH is to be underestimated. However, the PBLH is reproduced well by the model, and the model does not underestimate the PBLH (Fig. 7). The new scheme does not disturb the simulation results of meteorological fields, and therefore does not affect the simulation results of PBLH (Fig. S5). The results of the simulation of pollutant concentration are improved under the similar PBLH,

which further demonstrates that the simulation of pollutant concentration is not only controlled by the PBLH.



## 5.3 Influence of other processes

Overestimating of pollutant concentrations has been improved in Eastern China, but there are also some sites in northern China where pollutant concentrations are underestimated. These sites (i.e., Hebei and

Beijing) are mostly located in the east of the Taihang Mountains and the south of the Yan Mountains (Fig. 8). For example, in December 2016, the pollutant concentrations of all sites in Beijing were not underestimated. Jia et al. (2021) have found that the pollutant concentrations of two sites located in the south of Beijng (i.e., blue dots in Fig. S2 in Jia et al., 2021) are well reproduced by the model (i.e., away from the mountain). This phenomenon also occurred in 2013-2017 (Fig. 8), and the pollutant

concentrations were significantly underestimated at some sites (i.e., near the mountain). The boundaries of overestimated and underestimated sites are pronounced in Beijing-Tianjin-Hebei region (white dashed in Fig. 8), and the pollutant concentration is overestimated at some sites, which are away from the mountain (i.e., Tianjin and southeast of Hebei). Furthermore, we found that the TDC of particles in the new scheme is significantly smaller than the TDC of heat in the original scheme in the mountain area

(red rectangle in Fig. S6). The terrain will disturb the turbulence fields, making the stable stratification weakly stable/unstable. Theoretically, the reduced TDC will increase the pollutant concentration near the mountain, and improve the underestimation of pollutant concentration of the original scheme. However, the change of TDC does not improve the underestimation of pollutant concentration in the mountain area, which shows that the impact of other processes is more obvious in the mountain area. For instance, the

advection process is strongly related to the wind and pollutant concentration gradients from upwind areas to downwind areas (Gao et al., 2018). Figure S7 shows that the wind speed is much more overestimated in the mountain areas (two purple rectangles in Fig. S7i-l), but only the pollutant concentration is always underestimated in the BTH region (Fig. 3i-l). Clear gradients of wind speed and pollutant concentration exist in the BTH region (Fig. 3a-d; Fig. S7a-d), so these sites (i.e., closer to the mountain) may be

significantly affected by the advection process. Hence, the influences of other processes or topography in the mountain area deserve further consideration in the future.

Whether the simulation of chemical components has been improved, it cannot be well verified because of the lack of observational data. Although the observational components of $PM_{2.5}$ are not available to evaluate the simulation results of new scheme, CO, as a representative of primary pollutants, can be



compared to the observations. Results from new scheme with TDC of particles are more consistent with the observations than the original scheme (Fig. S8), which supports the improvement of $PM_{2.5}$ concentration (Fig. 5 and S8).

## 6 Conclusions and prospects

Mesoscale model faces numerous challenges during the heavy pollution events. One of these challenges

is the correct description of the turbulent mixing of pollutants. Although the model can reproduce the evolution of pollutants, the simulation of the diurnal variation of pollutants is fundamentally flawed, especially at night. Errors in estimation of pollutant concentration are primarily caused by defects in the turbulent mixing of pollutants in the model. Actually, there is a difference between the turbulent transport of heat and particles. This result inspires us to deal with the turbulent diffusion of heat and particles

separately. Therefore, based on the turbulent diffusion expression of particles proposed by Jia et al., 2021, we again audited the improvement of pollutant concentration in winter from 2013 to 2017.

The original scheme overestimates the surface $PM_{2.5}$ concentration by 11.8% (2013), 48% (2014), 23.8% (2015) and 20.9% (2017) at night, respectively. The new scheme has improved the overestimation of the surface $PM_{2.5}$ concentration in eastern China at night, and the average absolute bias of average value can

be reduced by 8.3% (2013), 17% (2014), 11% (2015) and 11.7% (2017), respectively. In the vertical direction, the pollutant concentration is decreased in the surface-layer while it increased in the upper layer. Moreover, the improvement of the pollutant concentration field does not interfere with changing meteorological fields. Although the PBLH affects the diffusion of pollutants, the simulation of pollutant concentration is not specifically controlled by the PBLH, but by TDC. However, TDC has a negligible

impact on the simulation of pollutant concentration at some sites with complex topography. $PM_{2.5}$ components cannot be used to evaluate the results of the simulation of the new scheme, due to the lack of observational data. CO, however, as a representative of primary pollutants, can be compared to observations. Results from new scheme are more consistent with the observations than the original scheme, which supports the improvement of $PM_{2.5}$ concentration. The new scheme could provide

promising guidance during the heavy pollution events. The turbulent transport mechanism and scalar parameterization is a complex topic (Smedman et al., 2007, Lemon et al., 2019; Couvreux et al., 2020;





Edwards et al., 2020), and beyond that, other processes also need in-depth understanding and exploration (Seinfeld et al., 2016; Shao et al., 2019; Emerson et al., 2020). Therefore, more research during the heavy pollution events, especially on the experimental side (e.g., extensive measurement campaigns), might

shed more light on the turbulent mixing process of pollutants and their mechanisms.

## Data availability

The surface $PM_{2.5}$ concentration, meteorological data, turbulent datasets and turbulent flux data of $PM_{2.5}$ are available by request (xiaoye@cma.gov.cn).

## Author contributions

Development of the ideas and concepts behind this work was performed by all the authors. Model execution, data analysis and paper preparation were performed by WJ and XZ with feedback and advice.

## Competing interests

The authors declare that they have no conflict of interest.

## Acknowledgments

The authors would like to acknowledge the Tsinghua University for the support of emission data.

## Financial support

This research is supported by the NSFC Project (U19A2044); National Key Project of MOST (2016YFC0203306); Atmospheric Pollution Control of the Prime Minister Fund (DQGG0104); Key Projects of Fundamental Scientific Research Fund of CAMS (2017Z001).

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

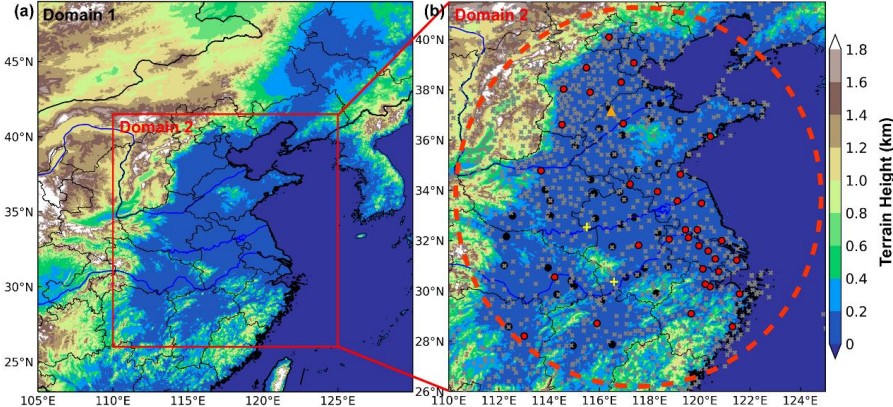

**Figure 1. (a) Map of terrain height in the two nested model domains. (b) The locations of surface**
**meteorological stations, air quality monitoring stations and sounding stations are marked by the**
**gray crosses, red(black) dots and yellow pluses, respectively. The turbulence data site is denoted by**
**the orange triangle. The red dashed circle indicates the areas of our primary concern.**



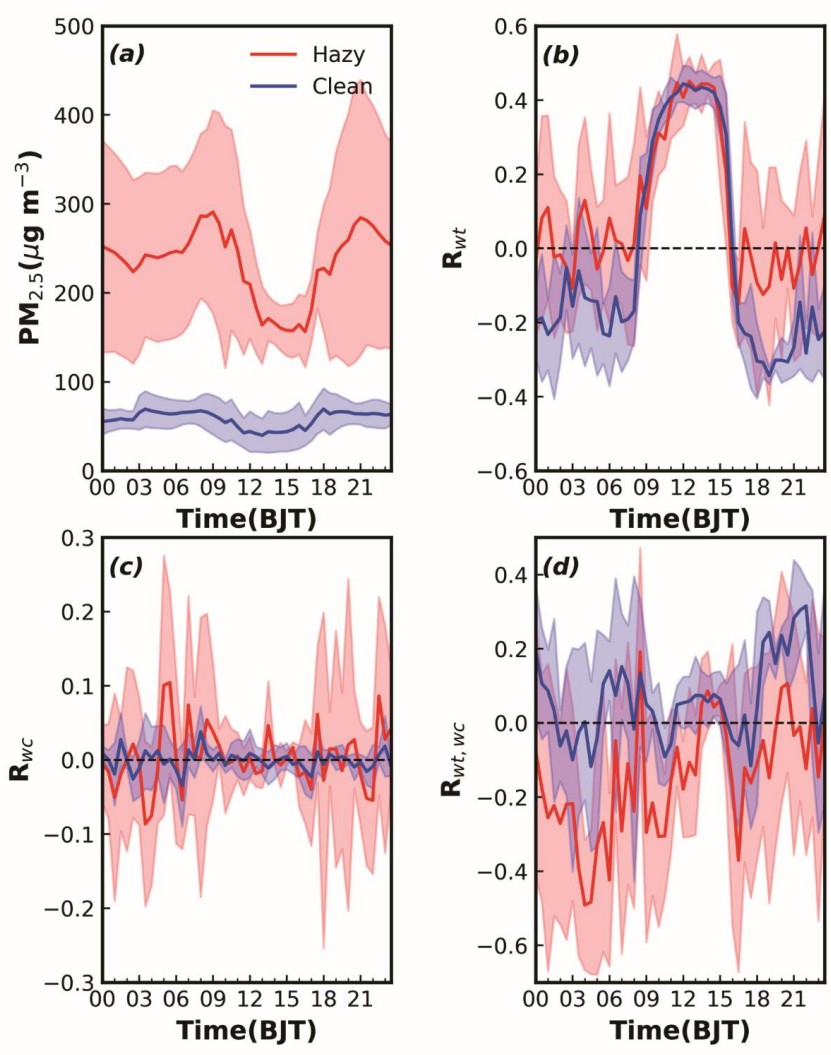

**Figure 2. Diurnal variations of (a) PM$_{2.5}$ concentration, (b) transport efficient of heat flux ($R_{wt}$), (c) transport efficient for fine particles flux ($R_{wc}$) and (d) correlations of heat and fine particles fluxes ($R_{wt,wc}$). The red (blue) line represents the heavy (cleaning) pollution episodes, and the shaded area indicates the average value ± standard deviation.**


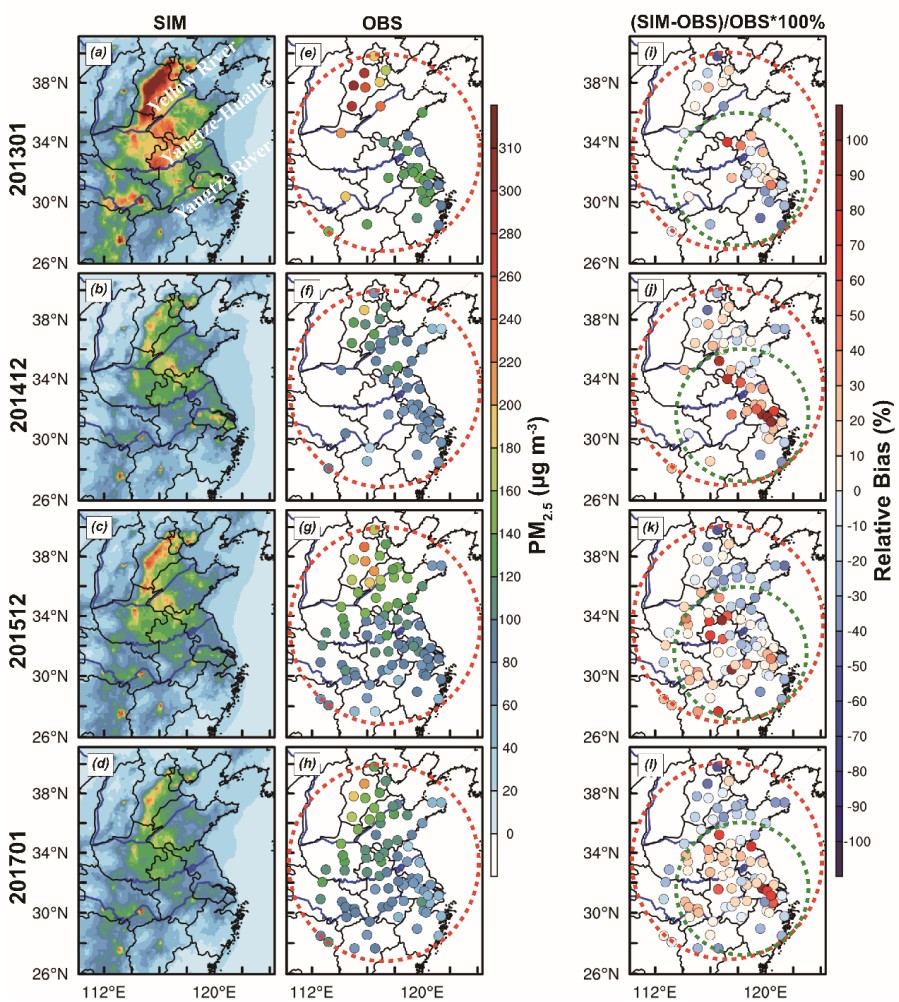

**Figure 3. The average value of (a-d) simulated and (e-h) observed PM₂.₅ concentration (μg m⁻³) at night, (i-l) the relative bias (RB, %) between simulation and observation, and the calculation formula of relative bias is** $RB = \left( \overline{X_{sim}} - \overline{X_{obs}} \right) \Big/ \overline{X_{obs}} \times 100\%$ **, where** $\overline{X_{sim}}$ **and** $\overline{X_{obs}}$ **represent the average value of simulation and observation, respectively. The locations of three rivers (i.e., Yellow River, Yangtze-Huaihe and Yangtze River) are marked by blue lines.**

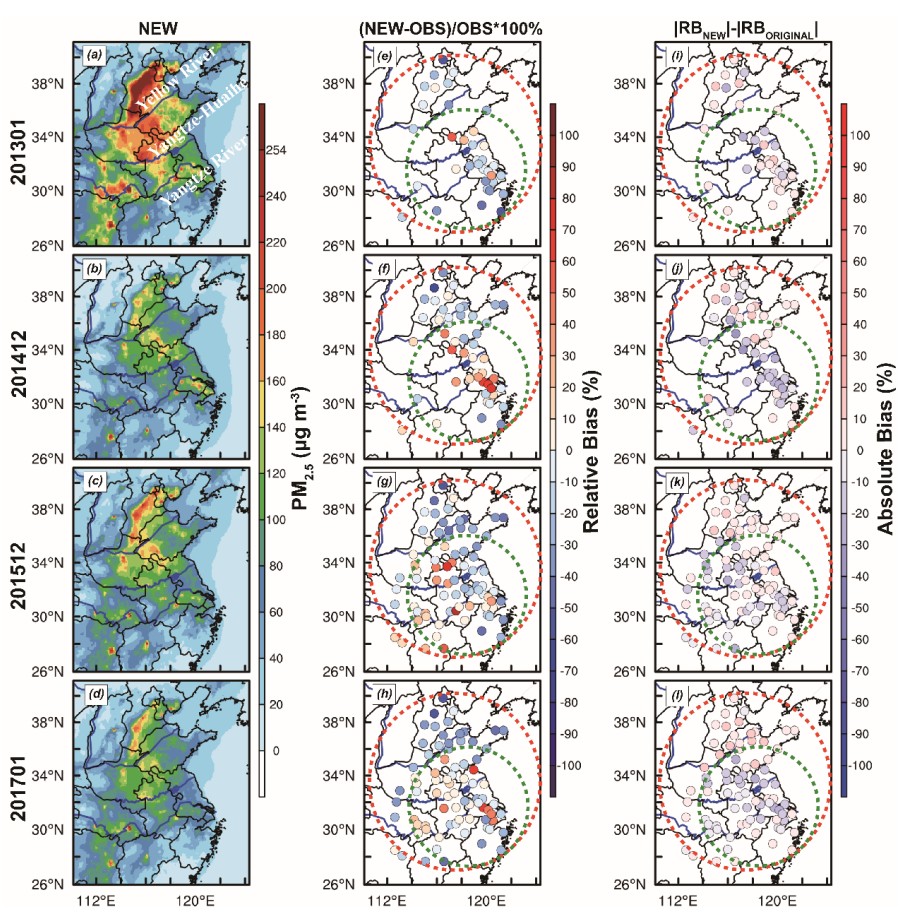

**Figure 4.** The average value of (a-d) simulated PM₂.₅ concentration (µg m⁻³) by new schemes, (e-h) the relative bias (RB, %) of PM₂.₅ concentration between simulation of new scheme and observation, (i-l) the absolute bias (AB, %) between new and original schemes, and the calculation formula of absolute bias is $AB = \left| RB_{new} \right| - \left| RB_{original} \right|$, where $\left| RB_{new} \right|$ and $\left| RB_{original} \right|$ represent the relative bias of new and original schemes, respectively.





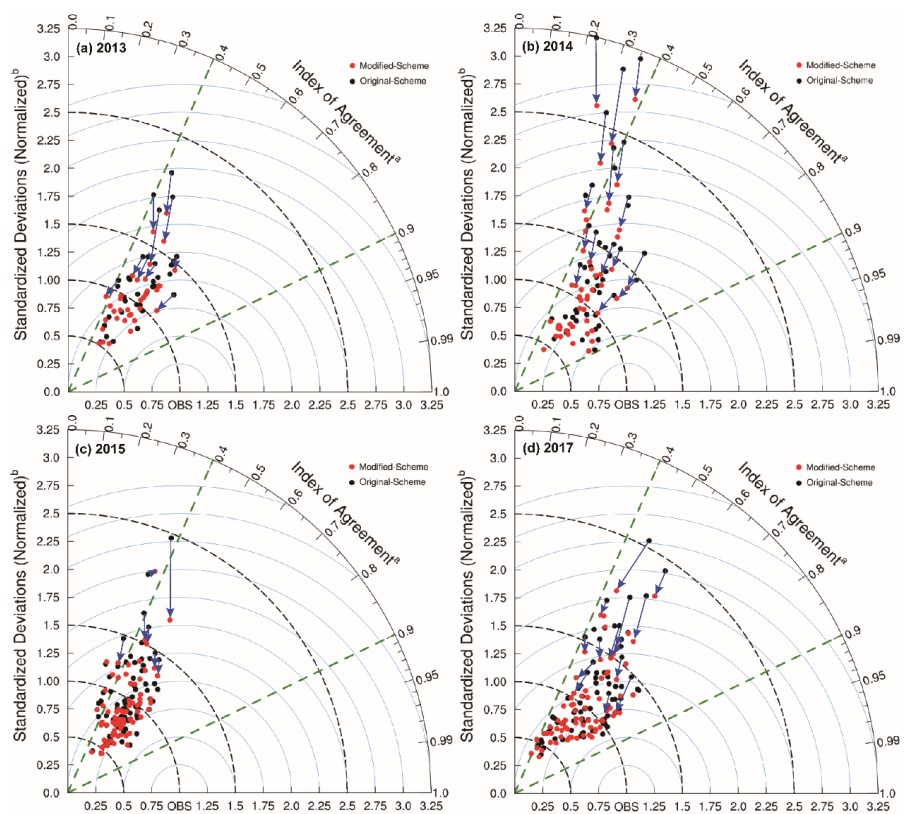

**Figure 5. Taylor diagram of simulation by original scheme and modified scheme. XY axes and arc represent the normalized standardized deviations (NSTD,**

$$NSTD = \frac{\sqrt{\dfrac{1}{N-1}\sum_{i=1}^{n}\left(X_{sim,i}-\overline{X_{sim}}\right)^2}}{\sqrt{\dfrac{1}{N-1}\sum_{i=1}^{n}\left(X_{obs,i}-\overline{X_{obs}}\right)^2}}$$ , $\overline{X_{sim}}$ **and** $\overline{X_{obs}}$ **represent the average value of**

**simulation and observation, respectively) and index of agreement (IOA,**

$$IOA = 1-\frac{\left[\sum_{i=1}^{n}|X_{sim,i}-X_{obs,i}|^2\right]}{\left[\sum_{i=1}^{n}(|X_{sim,i}-\overline{X_{obs}}|+|X_{obs,i}-\overline{X_{obs}}|)^2\right]}$$ , $X_{sim,i}$ **and** $X_{obs,i}$ **represent the value of**

**simulated and observed, respectively. i refers to time and n is the total number of time series), respectively. All cities (a total of 35 cities in 2013 and 78 cities in 2014, 2015 and 2017) are shown through dots, and black (red) represents original (new) scheme. The root mean square (RMS) is denoted by blue dashed line and the arrow indicates the change of the new scheme compared to the original scheme at the same station.**





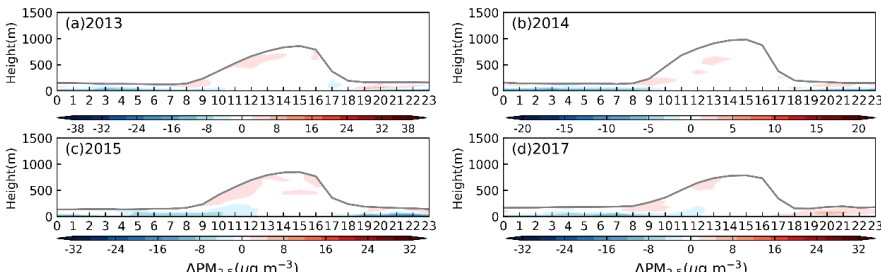

**Figure 6. Time-height cross sections for the difference of PM₂.₅ concentration between original and new schemes (i.e., the new scheme minus the original scheme) within the PBL. The gray line indicates the PBLH.**

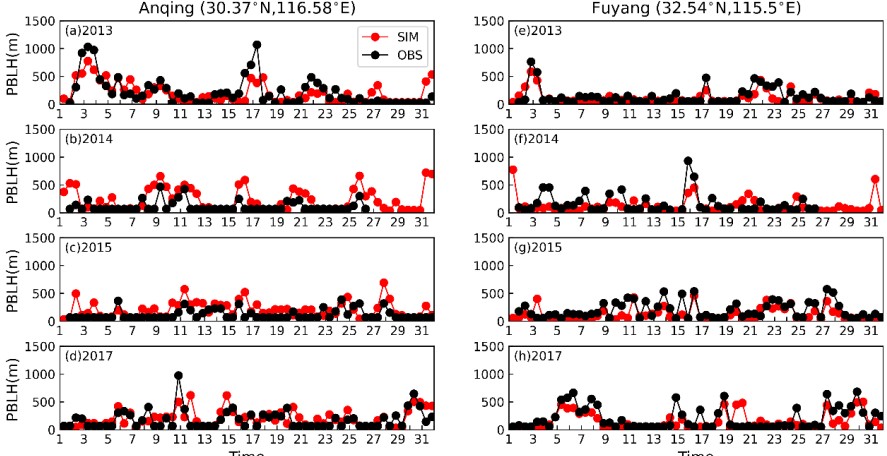

**Figure 7. Time series of the observed (black) and simulated (red) PBLH at 0800 and 2000 (BJT) in the (a-d) Anqing and (e-h) Fuyang from 2013 to 2017.**

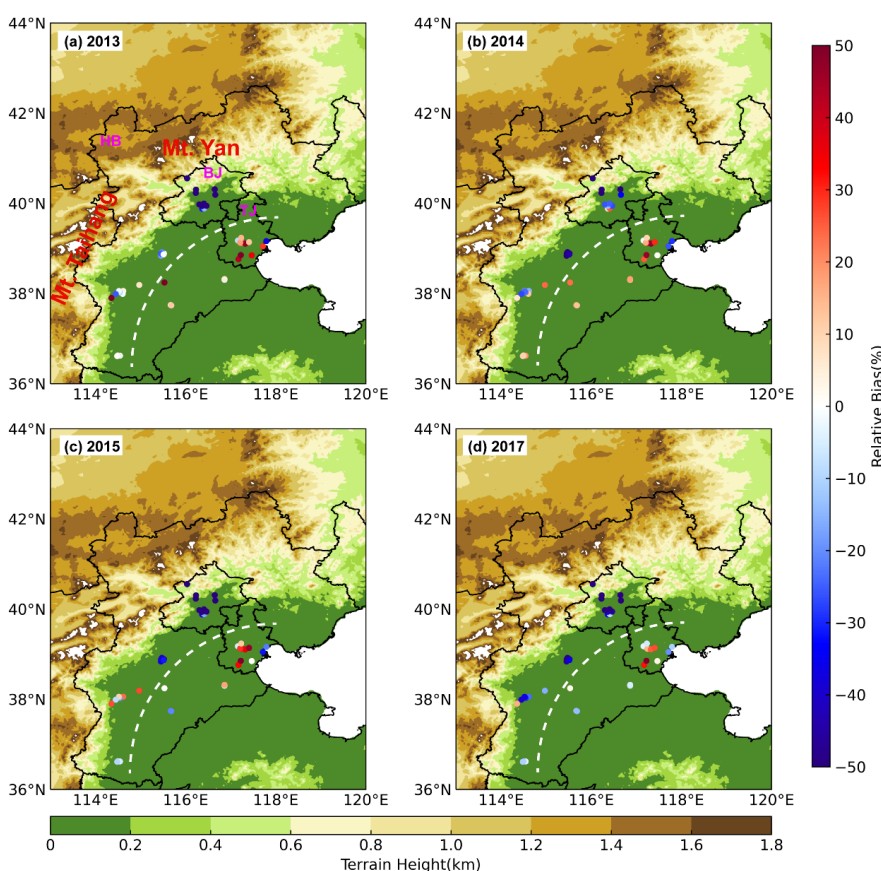

**Figure 8. The relative bias (%) between simulation and observation at all environment monitoring stations and terrain height in Beijing-Tianjin-Hebei in (a) 2013, (b) 2014, (c) 2015 and (d) 2017. Taihang mountain and Yan mountain are indicated by red text, Beijing (BJ), Tianjin (TJ) and Hebei (HB) are represented by purple abbreviation and the dividing line between overestimated and underestimated areas is indicated by a white dashed line.**