# Peer review of "diffusion process is insensitive to the simulation of the pollutant concentration. For mountainous area, the evolution of pollutants is more susceptible to advection transport. In addition to the PM2.5 concentration, CO as a primary pollutant, its concentration has also been improved,"

_Atmospheric Chemistry and Physics, 2021_

## Referee Comment (RC1)

Referee report for acp-2021-435

General Comments:

The scheme of turbulent mixing process of particles in the atmospheric boundary layer directly influences the particle concentration predicted by the numerical models. In this paper, the authors introduce a new scheme of turbulent diffusion coefficient, which is different from that for scalars, to describe the turbulent mixing process of particles in WRF-Chem model. The new scheme is only for stable conditions, while under unstable conditions the scheme is not changed. The results show that the new scheme can improve the prediction of particle mass concentration when compared to the output of model using the original scheme. However, the physics behind the results seems problematic. The numerical simulations show that the model using the new scheme reduces the overestimated PM2.5 concentration simulated using the original scheme. But the new scheme has a smaller turbulent diffusion coefficient (TDC) than the original one. It means that the model using the new scheme should predict larger PM2.5 concentration than using the original scheme. I cannot understand why a smaller TDC will lead to smaller PM2.5 concentration in the stable atmospheric boundary layer. So I think the authors should make additional analysis on their simulation results and tell us why a smaller TDC can result in smaller PM2.5 concentration under stable conditions (I mean, the authors should tell us the real reason for the reduced PM2.5 concentration simulated using the new scheme). Additionally, in this paper the authors provide some evidence to argue that the new scheme is reasonable for describing the turbulent mixing process of particles. I think the evidence is not convincing. In Section 3 the analyses of observational data show that the behaviors of transport efficiencies for heat and particle are different. But this result does not support the use of a smaller TDC for turbulent mixing of particle. As for the discussions in Section 5, the provided evidence does not tell us the physics that the model using the new scheme can have the better performance in predicting the PM2.5 concentration under stable conditions. So I think the presentation of this paper is not well done and the conclusions are unconvincing. My recommendation is major revision.

Specific Comments:

1) One of my major concerns is why the model using a smaller TDC can predict smaller PM2.5 concentration under stable conditions. Eqs. (2)–(5) give the new scheme of eddy diffusivities (the authors call eddy diffusivity as TDC) for heat, momentum and particle under stable conditions, in which the eddy diffusivity for particle $K_c$ is different to that for heat $K_h$. Actually, the difference between $K_c$ and $K_h$ is embodied by the different $Ri$-dependant functions, $f_c$ and $f_h$, as expressed in Eq. (5) and Eq. (3) respectively. The two equations indicate that in the new scheme $f_c$ is smaller than $f_h$ (in the original scheme $f_c$ is equal to $f_h$, as expressed in Eq. (3)). Therefore the new scheme has smaller eddy diffusivity for particle than the original scheme. To our knowledge, the smaller eddy diffusivity means the weaker turbulent transport, which results in higher concentration of air pollutant when other

conditions are the same. However, the simulations in this paper show that the model using the new scheme predicts smaller PM2.5 concentration than the model using the original scheme. My question is why. I think it is the most important issue in this paper. The authors should do additional analyses on the simulation results to find the reasons and tell us what the reasons are.

2) In Section 3, the authors present the analytical results of correlation coefficients $R_{wt}$, $R_{wc}$ and $R_{wt,wc}$ from observational data. They point out that the behaviors of turbulent transport of heat and particle are different, as shown in Figs. 2b–2d. Thus they use these results to support their choice of a different scheme of turbulent transport for particle from that for heat. In think these results do not make sense. On the one hand, Figs. 2b and 2c show that $R_{wt}$ and $R_{wc}$ are quite different under unstable conditions (i.e., in the daytime), but the authors choose the same scheme of turbulent transport for heat and particle. On the other hand, the behavior of $R_{wc}$ is not changed under both stable and unstable conditions—the values of $R_{wc}$ fluctuate around zero in the whole day as shown in Fig. 2c, but the authors choose different schemes of turbulent transport for particle under stable and unstable conditions respectively. As for the results show in Fig. 2d, I do not know what the results mean and how to interpret them. In my opinion, Section 3 is not a necessary part for this paper, because the results in this section cannot provide evidence to support the choice of a different scheme of turbulent transport for particle from that for heat under stable conditions. Actually, the authors choose the new scheme according to the results in Jia et al. (2021). They have cited the literature in this paper. That's enough. So I suggest the authors to delete this part, as well as Fig. S1.

3) For the results shown in Fig. 6, the authors state in lines 217-222 "Theoretically, increasing turbulent diffusion will reduce the pollutant concentrations near the surface-layer, and the pollutants will be more fully mixing in the vertical direction, which results in lower concentrations of pollutants in the near surface-layer and higher concentrations of pollutants in the upper layer. Actually, the pollutant concentration is reduced in the surface-layer and it is increased in the upper layer at night (Fig. 6), which is consistent with the theory". If the new scheme has a larger $f_c$, it is likely that these statements are reasonable. However, the new scheme actually has a smaller $f_c$. How to interpret the results shown in Fig. 6? Of course, a smaller $f_c$ unnecessarily means a smaller eddy diffusivity $K_c$. As expressed in Eq. (2), if $f_c$ becomes smaller, while the wind shear becomes much larger, we can still obtain a larger $K_c$. Does the model using the new scheme predict larger near-surface wind shear?

4) In subsections 5.1 and 5.2 the authors emphasize that the meteorological parameters and PBL height simulated by the new scheme are not changed when compared to those simulated by the original scheme (They provide evidence shown in Figs. S2–S5). They declare in lines 232-233 "noting that the new scheme does not alter the performance of meteorological fields, which is an advantage of the new scheme". They also declare in lines 264-266 "The results of the simulation of pollutant concentration are improved under the similar PBLH, which further demonstrates that the simulation of pollutant concentration is not only controlled by the

PBLH". I think these evidence cannot help us to know why the new scheme can reduce the overestimated PM2.5 concentration simulated by the original scheme. Given the unchanged meteorological field as well as PBLH, it seems that the reduced PM2.5 concentration can only be attributed to the new scheme of TDC. However, if the meteorological field and PBLH are really unchanged, the new scheme will have a smaller TDC and should predict larger PM2.5 concentration. So I suggest the authors to provide additional information about the simulation results. These information should tell us what are changed, as well as the relation between these changes and changed PM2.5 concentration.

5) Following the above comment, the same situation exists in subsection 5.3. I think the discussion about the influence of mountain terrain on the simulated PM2.5 concentration also cannot help us to know why the new scheme can reduce the overestimated PM2.5 concentration simulated by the original scheme. Furthermore, I do not understand the purpose of presenting the CO results in this subsection. Is the TDC for CO the same as that for particle in the new scheme? If not, why can the CO concentration be improved by the new scheme?

I have to say that the paper should be revised substantially. So I think there is no need to give the technical comments. Compared to my major concerns listed above, the technical problems are not important in the present stage.

---

## Author Comment (AC1)

**Response to Referee #2**

**General Comments:**

The scheme of turbulent mixing process of particles in the atmospheric boundary layer directly influences the particle concentration predicted by the numerical models. In this paper, the authors introduce a new scheme of turbulent diffusion coefficient, which is different from that for scalars, to describe the turbulent mixing process of particles in WRF-Chem model. The new scheme is only for stable conditions, while under unstable conditions the scheme is not changed. The results show that the new scheme can improve the prediction of particle mass concentration when compared to the output of model using the original scheme. However, the physics behind the results seems problematic. The numerical simulations show that the model using the new scheme reduces the overestimated PM2.5 concentration simulated using the original scheme. But the new scheme has a smaller turbulent diffusion coefficient (TDC) than the original one. It means that the model using the new scheme should predict larger PM2.5 concentration than using the original scheme. I cannot understand why a smaller TDC will lead to smaller PM2.5 concentration in the stable atmospheric boundary layer. So I think the authors should make additional analysis on their simulation results and tell us why a smaller TDC can result in smaller PM2.5 concentration under stable conditions (I mean, the authors should tell us the real reason for the reduced PM2.5 concentration simulated using the new scheme). Additionally, in this paper the authors provide some evidence to argue that the new scheme is reasonable for describing the turbulent mixing process of particles. I think the evidence is not convincing. In Section 3 the analyses of observational data show that the behaviors of transport efficiencies for heat and particle are different. But this result does not support the use of a smaller TDC for turbulent mixing of particle. As for the discussions in Section 5, the provided evidence does not tell us the physics that the model using the new scheme can have the better performance in predicting the PM2.5 concentration under stable conditions. So I think the presentation of this paper is not well done and the conclusions are unconvincing. My recommendation is major revision.

**Response:**

We feel great thanks for your professional review work on our article. As you are concerned, there are several problems that need to be addressed. According to your nice suggestions, we have made extensive corrections to our pervious draft, and the detailed corrections are listed below. Actually, we want to make a point, which is also the most important point, that is, the turbulent diffusion coefficient (TDC) in the new scheme is larger than that in the original scheme, not smaller. Therefore, using the new TDC will lead to lower pollutant concentration in the stable boundary layer.

Specific Comments:

1) One of my major concerns is why the model using a smaller TDC can predict smaller PM2.5 concentration under stable conditions. Eqs. (2)–(5) give the new scheme of eddy diffusivities (the authors call eddy diffusivity as TDC) for heat, momentum and particle under stable conditions, in which the eddy diffusivity for particle  $K_c$  is different to that for heat  $K_h$ . Actually, the difference between  $K_c$  and  $K_h$  is embodied by the different Ridependant functions,  $f_c$  and  $f_h$ , as expressed in Eq. (5) and Eq. (3) respectively. The two equations indicate that in the new scheme  $f_c$  is smaller than  $f_h$  (in the original scheme  $f_c$  is equal to  $f_h$ , as expressed in Eq. (3)). Therefore the new scheme has smaller eddy diffusivity means the weaker turbulent transport, which results in higher concentration of air pollutant when other conditions are the same. However, the simulations in this paper show that the model using the new scheme predicts smaller PM2.5 concentration than the model using the original scheme. My question is why. I think it is the most important issue in this paper. The authors should do additional analyses on the simulation results to find the reasons and tell us what the reasons are.

**Response:**

Here, we would like to point out that the turbulent diffusion coefficient (TDC) of the new scheme is greater than that of the original scheme, which can dispel your doubts. We agree with you that the difference between  $K_c$  and  $K_h$  is embodied by the different

*Ri*-dependent functions. Firstly, it can be seen from Figure R1 that the TDC of particles is significantly greater than that of heat, especially in the eastern China. Then, we can also find that the new TDC in the mountain area (i.e., indicated by red dotted rectangle in Figure R1) is smaller than original TDC, because when the whole region is in stable stratification, the complex terrain of the mountain area will disturb the flow field and making the stratification in the mountain area more unstable. With the increase of instability, the new TDC will be smaller than that of the original scheme (Figure 3 in Jia et al., 2021, we mainly compare red  $(f_c(Ri))$  and  $black(f_h(Ri))$  two solid lines). However, compared with the original scheme, the trend of the new TDC is a very good phenomenon for us. We found that the pollutant concentration was not only overestimated in eastern China by the original scheme, but also underestimated in mountainous area of North China. what we expect is that the trend of the new TDC can not only improve the overestimation of pollutant concentration in eastern China, but also improve the underestimation of pollutant concentration in mountainous are of North China. Unfortunately, the results are not as good as we expected. The increasing turbulent diffusion significantly reduces the simulated pollutant concentration in eastern China, but eh decreasing turbulent diffusion does not increase the pollutant concentration in mountainous area of North China. Therefore, on the one hand, this study proves that the main reason for the improvement of pollutant concentration in eastern China is that the new TDC plays a critical role, and puts forward the advantages of the new TDC in the model. On the other hand, we try to explain why the new TDC in mountainous area is insensitive to the simulation of pollutant concentration. We found that other processes may affect the change of pollutant concentration, such as advection transport. The wind speed in mountainous area is relatively high, and the pollutant concentration in the downwind area will be significantly affected when there is an obvious pollutant concentration gradient in the upwind and downwind areas.

---

## Author Comment (AC2)

**Response to Referee #1**

**Question1:**

This is the latest in a series of papers on air quality and aerosol issues in China that these authors have been involved with. There are considerable similarities between this manuscript and material in the cited GRL paper, Jia et al (2021b). The basic idea is that turbulent diffusion of heat differs from diffusion of momentum, of other scalars, and of aerosol particles. This is not a new idea and is generally dealt with in terms of $\varphi$ functions of z/L, where L is the Obukhov length (- u*3/[k(g/θ)<w'θ'>]. Here u* is the friction velocity, k is the Karman constant, θ is potential temperature and <...> denotes a time or ensemble average. In the current paper Eq (1), for eddy diffusivities (TDC), includes a stability function f(Ri) which differs between heat, fh, momentum, fm and particles, fc. This could be analogous to $\varphi$M(z/L), $\varphi$H(z/L) differences in the Monin-Obukhov approach.

**Response1:**

We feel great thanks for your professional review work on our article. As you are concerned, there are several problems that need to be addressed. It is true that the turbulent diffusion of momentum, heat and particles are different, and this is not a new idea. Previous studies have to study turbulent diffusion of particles by Computational fluid dynamics (CFD) model (Derudi et al., 2014; Fiates et al., 2016; Longo et al., 2019), experiment (Altunbas et al., 2002; Flesch et al., 2002; Sofiev et al., 2009), Reynolds-averaged Navier-Stokes equation (RANS) approach (Sini et al., 1996; Gualtieri, et al., 2017) and other means. However, for the mesoscale model, especially for a two-way coupled atmospheric-chemistry mesoscale model (e.g., WRF-Chem and GRAPES_CUACE), few people pay special attention to the turbulent diffusion of particles. Just as what you said, the stability functions (i.e., $f_h$, $f_m$, $f_c$) is analogous to $\varphi(z/L)$ (i.e., $\varphi_m(z/L)$ and $\varphi_h(z/L)$). Nonetheless, the turbulent diffusion of particles in the current mesoscale model is expressed by turbulent diffusion of heat, which has some errors. Therefore, based on this idea, we first establish the turbulent diffusion

relationship of particles based on Mixing length theory by using observational data, and then apply it to the mesoscale model (Jia et al., 2021). In our last article, we focused on establishing the turbulent diffusion relationship of particles through the observation data, and then added it into the model, which was preliminarily verified only by the results of 2016. In this manuscript, we mainly analyze the turbulent diffusion of particles from the perspective of model. The long-term simulation results are used to verify the reliability of the previous results, and the existing uncertainties are analyzed to provide the basis for future work.

**Question2:**

Eq (1) also includes a constant, 0.01, without any explanation or specification of units. It also appears to be absent in Jia et al (2021b). Given that the mixing length expression used in Eq (1) does not include a roughness length, z0, then one interpretation could be that $0.01 = ku*z0$. The issue is then whether there should be different roughness lengths for momentum, heat and aerosol.

**Response2:**

Actually, 0.01 refers to the minimum value of turbulent diffusion coefficient (TDC) in the model. For detailed parameter setting, refer to lines 653-841 (i.e., Subroutine EDDYX) of Program (i.e., module_bl_acm.F in WRF-Chem v3.9.1). Here, we have taken partial screenshot for reference (Fig. R1), where EDYZ0=0.01.

Figure R1. Partial screenshot of program (i.e., module_bl_acm.F in WRF-Chem v3.9.1).

**Question3:**

The present paper, and Jia et al (2021b) only present K(Ri) relationships for Ri > 0 (stably stratified conditions, while the WRF-CHEM model is run for day and night situations. Although the focus is on night-time conditions, we need to know what is done when Ri < 0 ($\partial\theta/\partial z < 0$). Is fc = fh in those cases?

**Response3:**

We are very sorry that we did not clearly explain the situation under all conditions. We have described the calculation principle of turbulent diffusion of particles under stable and unstable conditions, and see section 2.3 for details.

**Question4:**

The authors claim (line 134) that Monin-Obukhov similarity theory (MOST) is inapplicable and later (line 150) that "If the MOST is applicable, it indicates the turbulent mechanisms of heat, water vapor and particles are the same,..." without substantiating that erroneous claim. MOST is based on the idea of a surface boundary-layer with fluxes of heat and momentum being approximately independent of height. It is widely used within the surface layers of models such as WRF and ECMWF models. Dimensionless velocity and temperature gradient functions, $\varphi M(z/L)$, $\varphi H(z/L)$, based on MOST (e.g. Garratt, 1992, Eq 3.33 a,b) can differ and counter the line 150 claim. Admittedly these are in the unstable, $Ri < 0$, $L < 0$ case but there is nothing inherent in MOST to say that they should be equal in stable conditions.

**Response4:**

After the reviewer's suggestions, we have deleted the content of this section (i.e., temperature-particles transport dissimilarity). The connection between this section and other contents in the text is not very good, which is a little abrupt here. According to your suggestions later, we have also modified the title of the article, and the contents of the article is more in line with your suggested title.

**Question5:**

Negative remarks about MOST, here and in Jia et al (2021b) are used to support diffusion models based on gradient Richardson number, $Ri$ (without ever defining it). The problem with diffusion coefficients based on $Ri$ $[= (g/\theta)\partial\theta/\partial z/[(\partial U/\partial z)2 + (\partial V/\partial z)2]$ is that velocity and temperature gradients have strong z variation, basically proportional to $1/(z + z0q)$, where $z0q$ is the roughness length appropriate to the quantity involved, close to the surface and finite difference calculations of gradients can be very unreliable.

Meanwhile L is constant in a constant flux layer. In deeper layers, the flux Richardson number ($Rf=(g/\theta)<w'\theta'>/(\partial U/\partial z + <v'w'>\partial V/\partial z)$) is widely used. For aerosol in surface layers, MOST and Buckingham's Pi theorem, could allow an additional dimensionless variable ws/u*, where ws is the gravitational settling velocity, and could lead to interesting results allowing for variation between quantities being diffused by turbulence. Many models account for this via a deposition velocity for aerosol which combines the effects of turbulent diffusion and gravitational settling. The formulations of Zhang et al (2001) are a good example. Farmer et al (2021) show that deposition velocities, for micron sized particles, can vary significantly with particle diameter, underlying surface and friction velocity, and that "our understanding ... is poor".

**Response5:**

We are sorry that some concepts have not been clearly stated, and we have revised them. In fact, we have not made negative comments on MOST. In addition, previous studies have shown that the inapplicability of the MOST in the stable boundary layer (Edwards et al., 2020; Grachev et al., 2012), and our method is to avoid using the MOST under stable conditions. Moreover, we also evaluated the uncertainty difference between the two methods in the previous paper (Jia et al., 2021). We mentioned in our previous article that the TDC calculated by MOST and PBL height under stable conditions is uncertain, so we use the Mixing length theory to replace it. While under the unstable conditions, we still use MOST to calculate the TDC. Therefore, MOST and Mixing length theory are used in the model at the same time. We quite agree with you on the effects of gravitational settling, as Zhang et al. (2001) said, the parameterization of particle dry deposition is also extremely important. With increasing particle size, particle inertia and gravity cannot be neglected, but these inertia and gravity effects are neglected for particles smaller than 10 μm in diameter (Fratini et al., 2007). Therefore, we do not consider the gravity effect of particles here, but we added discussions on gravitational settling. In the future, we will use long-term simulation results to verify the difference of aerosol process decomposition in detail.

**Question6:**

An addition relative to Jia et al (2021a) are some data on correlation coefficients (Fig 2). It was not clear exactly what these data were averages of but from Ren et al (2020) we can find some details, which should be provided here. We should be told at what height these flux measurements are from. On average Rwt has a strong diurnal cycle while Rwc has a mean close to 0 implying minimal vertical flux. I assume that Rwc > 0 implies an upward flux of aerosol. Since much of the discussion is in terms of PM2.5 "pollution" and (line 95) gives information on anthropogenic emissions I had been thinking in industrial emission terms rather than land surface dust as the major component of the aerosol. Some clarification on this would be helpful.

**Response6:**

We have deleted this section. We hope we can have a separate article to study the turbulent transport between momentum, heat and particles in more detail based on the observational data (this work is in progress).

**Question7:**

Winter 2013-2017 Eastern China runs with the modified diffusion formulation for stable stratification are also new. We are told that PM2.5 concentration predictions are reduced. We are not really told why or where the PM2.5 particles go? Is the dust source reduced? Does more PM2.5 deposit on the ground, mix higher in the boundary layer or spread more widely in the horizontal? We are told nothing about deposition velocities but my guess would be that they average to zero (some + and some -) since Fig 2c shows near zero Rwc values.

**Response7:**

In fact, we have explained in Figure 5 that the pollutant concentration was reduced in the surface layer, and it was mixed to the upper level, and the pollutant concentration increases in the upper level. Firstly, there is no change in emission sources, so the impact of emission sources can be excluded. Secondly, the pollutant concentration

decreases near the surface layer, so more pollutants do not deposit on the ground. At the same time, the pollutant concentration increases in the upper level, and it is mixed in the boundary layer. Finally, if the pollutants are transported in the horizontal direction, when the underestimation of pollutant concentration at a certain station is improved, there will be no unified change driven by the winds in the Eastern China. Therefore, the pollutants are better mixed in the boundary layer.

**Question8:**

Overall this is a scientifically weak paper. It is not well written and has a strange title. That being said it is on an appropriate topic for ACP, it has some new results, relative to Jia et al, 2021b, although the basic idea and much of the discussion is similar. With Major Revision, less background material and fewer unnecessary references, plus the addition of some missing details, on Ri < 0, PM2.5 sources and sinks, surface boundary conditions, plus modelled aerosol budgets, then it could be publishable.

**Response8:**

Compared with the previous article (Jia et al., 2021), some of the results may be a little similar, mainly because we use the long-term simulation to verify the previous result, which are consistent with the previous results. In comparison, the previous article pays more attention to establish the turbulent diffusion relationship of particles based on the observational data. While this study pays more attention to the uncertainty analysis of the model results. According to your nice suggestions, we have made extensive corrections to our manuscript.

**Question9:**

As I see it, a mote appropriate title could be "Impact of modified turbulent diffusion of PM2.5 aerosol in WRF-Chem simulations in Eastern China". I cannot see that the manuscript demonstrates that a "Unified treatment of scalars is a missing source of turbulent diffusion on PM2.5 concentration in WRF-Chem".

**Response9:**

We have revised the title according to your suggestion.

References

Farmer, D.K., Boedicker, E.K. and DeBolt, H.M.: Dry Deposition of Atmospheric Aerosols: Approaches, Observations, and Mechanisms, Annu. Rev. Phys. Chem. 72:16.1–16.23, 2021.

Garratt, J.R.: The atmospheric boundary layer, Cambridge university Press, UK, 1992.

Zhang,.L, Gong, S., Padro, J., Barrie, L.: A size-segregated particle dry deposition scheme for an atmospheric aerosol module, .Atmos. Environ. 35:549–560, 2001.

**References**

Altunbas, A., Kelbaliyev, G., and Ceylan, K.: Eddy diffusivity of particles in turbulent flow in rough channels. J. Aerosol Sci., 33, 1075-1086, 2002.

Derudi, M., Bovolenta, D., Busini, V., and Rota, R.: Heavy gas dispersion in presence of large obstacles: selection of modeling tools, Ind. Eng. Chem. Res., 53, 9303–9310, 2014.

Edwards, J. M., Beijaars, A. C., Holtslag, A. A., and Lock, A. P.: Representation of boundary-layer processes in numerical weather prediction and climate models, Bound.-Lay. Meteorol., 177, 511–539, 2020.

Farmer, D. K., Boedicker, E. K. and DeBolt, H. M.: Dry Deposition of Atmospheric Aerosols: Approaches, Observations, and Mechanisms, Annu. Rev. Phys. Chem. 72, 16.1–16.23, 2021.

Fiates, J., Santos, R. R. C., Neto, F. F., Francesconi, A. Z., Simoes, V., and Vianna, S. S. V.: An alternative CFD tool for gas dispersion modelling of heavy gas, J. Loss Prevent. Proc., 44, 583-593, 2016.

Flesch, T. K., Prueger, J. H., and Hatfield, J. L.: Turbulent Schmidt number from a tracer experiment. Agr. Forest Meteorol., 111, 299-307, 2002.

Fratini, G., Ciccioli, P., Febo, A., Forgione, A., and Valentini, R.: Size-segregated fluxes of mineral dust from a desert area of northern China by eddy covariance. Atmos. Chem. Phys., 7, 2839–2854, 2007.

Grachev, A. A., Andreas, E. L., Fairall, C. W., Guest, P. S., and Persson, P. O. G.: Outlier problem in evaluating similarity functions in the stable atmospheric boundary layer. Bound.-Lay. Meteorol., 144, 137–155, 2012.

Gualtieri, V, Angeloudis, A., Bombardelli, F., Jha, S., and Stoesser, T.: On the Values for the Turbulent Schmidt Number in Environmental Flows. Fluids, 2, 17, 2017.

Jia, W., Zhang, X., Zhang, H., and Ren, Y.: Application of turbulent diffusion term of aerosols in mesoscale model, Geophys. Res. Lett., 48, e2021GL093199, https://doi.org/10.1029/2021GL093199, 2021.

Longo, R., Furst, M., Bellemans, A., Ferrarotti, M., Derudi, M., and Parente, A.: CFD dispersion study based on a variable Schmidt formulation for flows around different configurations of ground-mounted buildings, Building and Environment, 154, 336-347, 2019.

Sini, J.-F., Anquetin, S., and Mestayer, P. G.: Pollutant dispersion and thermal effects in urban street canyons. Atmospheric Environment, 30, 2659–2677, 1996.

Sofiev, M., Sofieva, V., Elperin, T., Kleeorin, N., Rogachevskii, I., and Zilitinkevich, S. S.: Turbulent diffusion and turbulent thermal diffusion of aerosols in stratified atmospheric flows. J. Geophys. Res., 114, D18209, 2009.

Zhang,.L, Gong, S., Padro, J., and Barrie, L.: A size-segregated particle dry deposition scheme for an atmospheric aerosol module. Atmos. Environ., 35, 549–560, 2001.

---

## Referee Report (RR1)

RE: Impact of modified turbulent diffusion of PM2.5 aerosol in WRF-Chem simulations in Eastern China
Author(s): Wenxing Jia and Xiaoye Zhang
MS No.: acp-2021-435: MS type: Research article; Iteration: Revised submission

General Comments - Peter Taylor

The authors have changed the title, as suggested, and made a series of changes but I still have concerns about some missing details. There are also sections, including the abstract, where some careful language edits could clarify the text.

The Richardson number dependent eddy diffusivity (Eq2, 5) for particles in stable conditions ($Ri > 0$) was developed and discussed in Jia et al (2021b). In that paper Ri is said to be a **gradient** Richardson number "where $f(Ri)$ is empirical stability function of_ gradient _Richardson number ($Ri$)." but **is not defined**. The definition given in the present paper (Eq 1) is a bulk Richardson number based on differences relative to "surface level values". For winds are these 0 or $U_{10}$? Tracking down Ri use by different authors is always a problem. Esau and Byrkjedal (2007) use finite difference approximations to the local gradient Ri, and discuss issues associated with their accuracy, " Another important aspect of the vertical resolution is a numerical approximation of strongly-curved vertical profiles on coarse meshes using finite-difference numerical schemes." The current paper seems to use Esau and Byrkjedal's TDC functions for heat and momentum, but with a bulk $Ri$.

Fig 3 of Jia et al (2021b) shows the data used to develop Eq 5. There is a lot of scatter, near zero values of $f_c$ for several data points with $Ri < 0.2$. For $Ri > 0.2$ there are some large (~0.6) experimental $f_c$ values. Bottom line is that I would have very little confidence in the $f_c$ formulation proposed by Jia et al (2021b).

If implementation in WRF-CHEM as a tuning exercise produced well documented, convincing results that could certainly be of interest but I am not convinced by the material here. We are given information on the impacts of changes to the TDC for particles on model results. They do appear to improve overall comparisons with observed average values but I would like to see much more detailed discussion. One set of daily PBL height values (Fig 6) are given as a time series but it would be informative to see more sample time series data, of $PM_{2.5}$ concentrations and fluxes at observation points, and with hourly resolution to see day-night differences. Start with some time series comparisons and then worry about the overall statistics. It is hard to take output from a large, complex meteorological model like WRF-CHEM in order to see the impact of tuning one of the internal equations but we should be shown more of the details.

The confusion about exactly what is used as $Ri$ is concerning. Does the model $Ri$ correspond to the value used from the measurement data from Jia et al (2021b)?

Many of my concerns relate to that previous GRL paper but it would not be good for ACP to further encourage the use of a dubious result.

I am also concerned about the sources of $PM_{2.5}$. We are simply told to believe their inventory. Are we looking at $PM_{2.5}$ from smoke stacks and subsequent chemical transformation, or is some of it road and other dust? Is the ground surface always a sink or can it be a source? How can we be sure that the differences between model prediction and observations are not related to the source inventory?

Detailed comments

line 105    Were these PM2.5 flux measurements compared with model predictions?

line 114    Good to calculate observed vertical fluxes, but are the results used or compared with the modelled fluxes. Is the ground a source or sink of $PM_{2.5}$ in these simulations?

line 123    Note that Eq(1) is a bulk Ri, and explain what are the surface level values, $u_s$, $v_s$, $\vartheta_{vs}$.

line 141    .. horizontal **grid** resolution....  and the factor 5 is rather larger than usually applied.

line 147    ? **Single** layer UCM , and explain UCM.  If measurement sites are urban then this could be an important issue.

line 165    A 64 h spin up seems long for WRF (typically 12 h for meteorology) but may be needed for WRF-CHEM if initial concentrations are unknown, also why 91?  64 + 24 = 88?  Can you explain reasons for the long spin up time. Are results sensitive to this?

line 175    Is the 0.01 minimum value common to the original and new schemes?

line 176    If one wants u = 0 at z =0 then one can set $l = k(z+z_0)/(1 + kz/\lambda)$. where $z_0$ is a roughness length. There is no mention of roughness length until line 353, but it can be an important parameter and should be different for momentum, heat and $PM_{2.5}$. It is present in WRF and should be discussed. It could be linked to the 0.01 minimum in Eq (2)?

line 186    It is worth comparing the current $f_h$ and $f_c$, with the expression one would get using MOST, if we use the standard form $1/f_h = \varphi_h = 1 + 5 \, z/L$, and relationship with Ri from Garratt (p52), and assuming L = 0 for Ri > 0.2.  A quick plot is below with $f_{most} = 1/ \varphi_h$.

[Figure]

For moderate values (<0.2) of *Ri*, MOST and $f_h$ are both significantly larger than $f_c$ so that the modified TDC is significantly reduced relative to MOST or earlier assumptions with $K_c = K_h$.

For Ri > 0.25, $f_c > f_h$ as noted on line 188, but how often does this occur - in reality and in the model. In the model there may be confusion between gradient and bulk Ri. Can we see a pdf of *Ri* values?

line 199    "avoids the inapplicability of MOST".  This "inapplicability" should be explained, maybe it is because we are not necessarily in a constant flux layer? MOST need not require $f_c = f_h$.

line 208    How is "night" defined in forming these averages?

line 223    Why the big difference in 2014? I am puzzled by why "relative bias" and "absolute bias" % differences are different. I assume that these are night-time values? Based on hourly data?

From Fig 3 the relative and absolute bias values seem to be of opposite sign at some locations, A detailed definition of these quantities should be provided. I can guess but a few equations would help.

line 235    So n in Fig 4 is 31*24?  I am not familiar with Taylor diagrams. Is the vertical axis from the model and the horizontal the observations?

line 261    Can you say anything about deposition to the surface, which could be a critical removal process, or a source?

line 287    There is a lot to be said about keeping models as simple as possible. What exactly do you have in mind as a " turbulence-aerosol two-way feedback module". But not needed here.

line 344    This seems to be the first mention of "source" and we are expected to accept that " there is no way to use other more elaborate inventories to quantify the uncertainty caused by emissions".

line 395    Maybe tell us what the mean absolute errors in hourly or daily $PM_{2.5}$ values ($\mu g\ m^{-3}$) are in order to see how significant these bias improvements are.

line 398    " Therefore, the pollutant concentration is reduced near the surface and better mixed in the whole layer, increasing the pollutant concentration in the upper level."  Are there any upper level measurement to validate this effect?
* * *
Reference.

Garratt, J.R., (1992) The atmospheric boundary layer, Cambridge, UK

---

## Referee Report (RR2)

RE:  Impact of modified turbulent diffusion of PM2.5 aerosol in WRF-Chem simulations in Eastern China
Author(s): Wenxing Jia and Xiaoye Zhang
MS No.: acp-2021-435: MS type: Research article; Iteration: Revised submission

General Comments - Peter Taylor

The authors have made a number of changes and have provided detailed responses to questions raised about earlier versions. I still have concerns about the modified stability function used, Eq 4, in connection with the Turbulent Diffusion Coefficient (TDC) and feel that the degree of improved performance is exaggerated.  Despite Response 14, I am not really sure whether the % improvements are C/A or C/OBS values. Providing the bias values of both the original and the new schemes might be clearer. From Fig 4, Relative Bias is given as (NEW-OBS)/OBS and it is clear that individual stations have a wide range of bias % and that modest improvements of order 10% (line 453) in the mean over all stations may not warrant the claim, line 286 that "the new scheme can significantly reduce the degree of overestimation", and similar on line291. Also line 452 could make it clear whether these are % reductions of % overestimates rather reduction in the % overestimates themselves? Reduced by, or reduced to?

While I appreciate the need to reference relevant prior work I am surprise that a paper on a relatively narrow topic needs about 60 references. The paper is well written but will need some language editing.

Minor points

p5 Line 109        It may be misleading to say that the roughness is considered as zero.  As the authors note in Response 8 to questions on the previous version, WRF does treat the PBL and surface layer and, unfortunately, the PBL code ignores $z_0$. Although no changes have been made to the surface layer code it probably does involve a roughness length based on land use maps. WRF boundary layer modules, MYNN and YSU make use of $z_0$ values based on land use. At this stage just avoid discussing $z_0$ unless you plan to dig into the WRF code and find out.

p5 line 120        Does this suggest that Ri values in the original field data were based on (60m - 10m) differences?  Do these really give representative "gradients"

p6 line 141        I did not have time to look into the "L-band radiosonde system" but assume it transmits 1Hz data as it rises. The issue is what vertical z resolution does this represent - what is the rise rate?

p7 line 185        Not clear what the 8 months are, 4 with the original model and 4 with the new?

p7 line 185        Re Fig 2, I am just curious whether kstart = 1 always in these runs?

p8 line 210 ++   It is fine to use fine grid finite differences to approximate gradient Ri. Formally a bulk $Ri_B$ would be based on two widely separated levels, one of which is normally the surface, and should be specific to those levels. As Garratt (1992, p37) notes, politely, some authors use the bulk term incorrectly.

p11 line 269      See general comment. Need to make clear what "mean absolute bias" means.  Is it the mean of an absolute value <|OBS-Model|> or just <OBS-Model>, where <...> means "mean".

p15 line 393      Are model values of dry deposition available? Do they play a significant role in the $PM_{2.5}$ budget?

---

## Author Response (AR2)

**Response to Editor**

The author sincerely thanks the editor for giving us an opportunity to revise the manuscript again, as well as the reviewers' professional evaluation and valuable suggestions. According to suggestions from reviewers, we have made extensive corrections to our previous manuscript, and the detailed point-by-point responses are listed below.

**Response to Referee #1**

**RE:** Impact of modified turbulent diffusion of PM2.5 aerosol in WRF-Chem simulations in Eastern China

**Author(s):** Wenxing Jia and Xiaoye Zhang

**MS No.:** acp-2021-435: MS type: Research article; Iteration: Revised submission

**General Comments** - Peter Taylor

**Question1:**

The authors have changed the title, as suggested, and made a series of changes but I still have concerns about some missing details. There are also sections, including the abstract, where some careful language edits could clarify the text.

**Response1:**

We feel great thanks for your professional review work on our article. As you are concerned, there are several problems that need to be addressed. According to your nice suggestions, we have made extensive corrections to our pervious manuscript, and the detailed point-by-point responses are listed below. We have reorganized the abstract, please see Lines 13-40 in the revised manuscript.

**Question2:**

The Richardson number dependent eddy diffusivity (Eq2, 5) for particles in stable conditions (Ri > 0) was developed and discussed in Jia et al (2021b). In that paper Ri is said to be a gradient Richardson number "where f(Ri) is empirical stability function of gradient Richardson number (Ri)." but is not defined. The definition given in the present paper (Eq 1) is a bulk Richardson number based on differences relative to "surface level values". For winds are these 0 or U10? Tracking down Ri use by different authors is always a problem. Esau and Byrkjedal (2007) use finite difference approximations to the local gradient Ri, and discuss issues associated with their accuracy, " Another important aspect of the vertical resolution is a numerical

approximation of strongly-curved vertical profiles on coarse meshes using finite-difference numerical schemes." The current paper seems to use Esau and Byrkjedal's TDC functions for heat and momentum, but with a bulk Ri.

**Response2:**

We are very sorry that the description of data and methods in Section 2 are not clear enough, and even puzzles you. Actually, we have been using gradient Richardson number in previous and current papers, and the gradient Richardson number is approximated in finite difference form and the resulting is sometimes referred to as the bulk Richardson number (Garratt, 1992). For example, Louis et al. (1982) shows the Ri is the bulk Richardson number, but the expression is the form of gradient Richardson number (Eq .(5) in Louis et al., 1982). Moreover, our calculation method is consistent with that in the model. Equation (1) introduces the calculation method of PBL height, which leads to many doubts. Because this method has been widely used (Guo et al., 2016; Miao et al., 2018; Seidel et al., 2012; Zhang et al., 2013), it has been deleted to avoid confusion in the revised manuscript. In the revised manuscript, we will reorganize the content of Section 2 and add a flow chart (Figure 2) to show more details. In the original ACM2 scheme (in WRFv3.9.1), the Easu and Byrkjedal's TDC functions are used to calculate heat and momentum, and the Ri in the model is calculated as follows:

$$DZF = ZA(i, k+1) - ZA(i, k) \qquad (1)$$

$$SS = \frac{\left(US(i, k+1) - US(i, k)\right)^2 + \left(VS(i, k+1) - VS(i, k)\right)^2}{DZF^2} \qquad (2)$$

$$GOTH = \frac{2G}{\theta_v(i, k+1) + \theta_v(i, k+1)} \qquad (3)$$

$$Ri = \frac{GOTH \cdot (\theta_v(i, k+1) - \theta_v(i, k+1))}{DZF \cdot SS} \qquad (4)$$

where *ZA* is the heigh of each level in the model, which is related to the setting of the model. To resolve the PBL structure, 21 vertical layers were set below 2 km in this study (i.e., the specific setting of vertical levels is σ= 1.000, 0.997, 0.994, 0.991, 0.988, 0.985, 0.980, 0.975, 0.970, 0.960, 0.950, 0.940, 0.930, 0.920, 0.910, 0.895, 0.880, 0.865, 0.850, 0.825, 0.800). *i* and *k* represent gird points in horizontal and vertical direction respectively. *US* and *VS* are the component of wind. G is the gravity, $\theta_v$ is the virtual potential temperature. Then, substitute equations (1)-(3) into the equation (4) to calculate the Ri. Finally, Ri is used to calculate the turbulent diffusion coefficient.

We also need to note that the version of the model is very important, and Easu and Byrkijedal's TDC functions are updated only in the version after WRFv3.6.1. The improvement shows the importance of turbulent diffusion in the PBL scheme.

Please see Section 2 for details in the revised manuscript.

**Question3:**

Fig 3 of Jia et al (2021b) shows the data used to develop Eq 5. There is a lot of scatter, near zero values of fc for several data points with Ri < 0.2. For Ri > 0.2 there are some large (~0.6) experimental fc values. Bottom line is that I would have very little confidence in the fc formulation proposed by Jia et al (2021b).

**Response3:**

We quite agree with you. We cannot rashly adopt a new functional form for fitting, because of the scarcity of data. However, these only data selected through field observation and strict quality control (Ren et al., 2020). Therefore, we still use a series of function forms proposed by predecessors for fitting, and finally select a better equation. For Ri>~0.2, there are some large experimental fc values, there are four points that do not follow the tail behavior, and account for 8% of the number of scattered points in the figure. We check these abnormal points, finding that these points correspond to a larger turbulent flux. We have also speculated whether the turbulent diffusion of particles may be larger than what is fitted now. Although the current

function coefficient needs to be discussed, the ideas and techniques are worthy of deliberation. After more observation data are obtained, we will have a more accurate coefficient. The change of function coefficient will only make our future results better, and the continuous development and progress of the model also needs everyone's joint efforts to improve step by step. We are also doing our best to get more observation data (in progress) and further improve our fitting function.

**Question4:**

If implementation in WRF-CHEM as a tuning exercise produced well documented, convincing results that could certainly be of interest but I am not convinced by the material here. We are given information on the impacts of changes to the TDC for particles on model results. They do appear to improve overall comparisons with observed average values but I would like to see much more detailed discussion. One set of daily PBL height values (Fig 6) are given as a time series but it would be informative to see more sample time series data, of PM2.5 concentrations and fluxes at observation points, and with hourly resolution to see day-night differences. Start with some time series comparisons and then worry about the overall statistics. It is hard to take output from a large, complex meteorological model like WRF-CHEM in order to see the impact of tuning one of the internal equations but we should be shown more of the details.

**Response4:**

In fact, numerous scholars have been debugging a parameter in the mesoscale model for a long time, so as to change the simulation results of meteorological parameters and pollutants. For example, numerical weather prediction (NWP) usually adopts stability functions with so-called "enhanced mixing" to improve the unrealistic surface cooling (Bejaars and Holtslag, 1991; Derbyshire, 1999). In addition, the simulation of meteorological parameters and pollution concentration can be improved by specifying a minimum background turbulent diffusion under the stable conditions (Du et al., 2020; Savijärvi and Kauhanen, 2002). For the ACM2 scheme itself, with the continuous

updating of the version, the turbulent diffusion coefficient is gradually improved. Because the turbulent diffusion coefficient is extremely important, it controls the vertical mixing of momentum, heat, water vapor and pollutants within the PBL. Therefore, changes in turbulent diffusion coefficient can significantly affect the change of meteorological parameters and aerosol pollutant concentration.

We have described the improvement of the PBL scheme in further detail in the Section 2 and Figure 2 in the revised manuscript. We have made the comparison of pollutant concentration time series in the early stage. Since we need to simulate for a long time and compare many stations, finally, we show the monthly average results of regional distribution. Here we take three typical cities (i.e., Hefei, Nanjing and Shanghai) in Eastern China as an example to show the comparison of simulated and observed pollutant concentration (Fig. R1).

It can be seen from Figure R1 that the new scheme significantly improves the pollutant concentration is overestimated during heavy pollution episodes. At the same time, for the underestimated periods, the new scheme does not further worsen the underestimated situation. This is also an advantage of the new scheme. There must be some difference in the simulation results of individual cases, because of the difference of pollution cases. In short, the overall average results are excepted (Fig. 5).

[Figure]

**Figure R1.** Time series of PM$_{2.5}$ concentration in Hefei, Nanjing and Shanghai from 2013 to 2017. The black, red and blue lines represent the results of observation, original scheme and scheme, respectively. The green dashed box indicates the period when the new scheme has significantly improved.

**Question5:**

The confusion about exactly what is used as Ri is concerning. Does the model Ri correspond to the value used from the measurement data from Jia et al (2021b)?

**Response5:**

The Ri we used is definitely the same as Jia et al (2021), and we can provide time series of Ri (Fig. R2).

[Figure]

**Figure R2.** Time series of Richardson number from 27 December 2018 to 8 January 2019.

**Question6:**

Many of my concerns relate to that previous GRL paper but it would not be good for ACP to further encourage the use of a dubious result.

I am also concerned about the sources of PM2.5. We are simply told to believe their inventory. Are we looking at PM2.5 from smoke stacks and subsequent chemical transformation, or is some of it road and other dust? Is the ground surface always a sink or can it be a source? How can we be sure that the differences between model prediction and observations are not related to the source inventory?

**Response6:**

We fully understand your concerns. In fact, the data used in this study have undergone strict quality control and many turbulence information characteristics have been

described in Ren et al (2020). Except that the sample size may be a little small. However, for the problem of sample size, we are also continuing to carry out field observation experiments, and we will obtain more data for our follow-up research in the future.

We very much agree with your question about the inventory. At present, the inventory we used is Multi-resolution Emission Inventory for China (MEIC) provided by Tsinghua University. The inventory published on the website has been updated to 2017 (http://meicmodel.org/), which is also the latest inventory published so far. Indeed, the accuracy of emission sources is very important for the simulation of pollutant concentration. In the WRF-Chem model, the processes of emission source, turbulent vertical mixing, dry deposition, advection transport and chemical conversion have a significant impact on the simulation of pollutants (Du et al., 2020; Chen et al., 2019; Gao et al., 2018). We also want to quantitatively prove the uncertainty of emission source, but this idea is difficult to realize at present. The focus of this study is to understand the impact of turbulent diffusion on pollutant concentration by adding turbulent diffusion of pollutants to the mesoscale model. It can be seen from the results of this paper that the simulated pollutant concentration is not completely consistent with the observation by improving the turbulent diffusion process. Therefore, the simulation results still have some room for improvement, and more efforts are needed in each process in the future.

**Detailed comments**

**Question1:**

line 105 Were these PM2.5 flux measurements compared with model predictions?

**Response1:**

These $PM_{2.5}$ flux measurements are used to calculate the Richardson function of particles, i.e., $f_c(Ri)$.

**Question2:**

line 114 Good to calculate observed vertical fluxes, but are the results used or compared with the modelled fluxes. Is the ground a source or sink of PM2.5 in these simulations?

**Response2:**

Vertical flux is used for the calculation of the function of Richardson. In all simulation cases, $PM_{2.5}$ may be both source and sink. We did not simulate a specific pollution process, but a long-time simulation. Therefore, the situation of each case is different. When there is a transport stage of a case, $PM_{2.5}$ is a sink, while if it is a stable stratification with small wind, $PM_{2.5}$ is more likely to be a source. If we want to know the source or sink of pollutants, we can only conduct simulation analysis for individual cases. In addition, the pollution process is complex. One pollution process is not only caused by source or sink, but is likely to have different results in different time periods (Zhong et al., 2018; 2019).

**Question3:**

line 123 Note that Eq(1) is a bulk Ri, and explain what are the surface level values, us, vs, θvs.

**Response3:**

We are sorry to bother you with this equation. This equation is a method for calculating the PBL height and has been widely used in models and observations (Guo et al., 2016; Miao et al., 2018; Seidel et al., 2012; Zhang et al., 2013). The surface level means the observation height is about 10 m (Miao et al., 2018). We have rewritten this section, please see Section 2 in the revised manuscript.

**Question4:**

line 141 .. horizontal grid resolution.... and the factor 5 is rather larger than usually applied.

**Response4:**

Since we focus on a large area, the simulation time is relatively long, and used a two-way coupled meteorological-chemistry model, we do not use other horizontal grid resolutions. This ratio is more suitable for our simulation process. Of course, the ratio of 3 and 5 have been used by scholars, and the nesting of different ratio has little effect on the simulation results of the original scheme. Because other scholars can also find the pollutant concentration is overestimated in Eastern China (Du et al., 2020; Wang et al., 2021).

**Question5:**

line 147 ? Single layer UCM , and explain UCM. If measurement sites are urban then this could be an important issue.

**Response5:**

There are three urban surface physical schemes, namely, UCM, BEP and BEM.

UCM: Urban canopy model: 3-category UCM option with surface effects for roofs, walls, and streets.

BEP: Building Environment Parameterization: Multi-layer urban canopy model that allows for buildings higher than the lowest model levels.

BEM: Building Energy Model. Adds to BEP, building energy budget with heating and cooling systems.

The reason why we use UCM scheme here is that UCM can match any PBL schemes. However, BEP and BEM schemes can only match MYJ and BouLac PBL schemes. Therefore, we chose the UCM schemes.

**Question6:**

line 165 A 64 h spin up seems long for WRF (typically 12 h for meteorology) but may be needed for WRF-CHEM if initial concentrations are unknown, also why 91? 64 + 24 = 88? Can you explain reasons for the long spin up time. Are results sensitive to this?

**Response6:**

Indeed, the spin-up time can be shorter in WRF, as you said, 12 hours. However, the variables have increased a lot in the WRF-Chem, and the simulation area is large. Therefore, we lengthened the spin-up time to obtain more stable calculation results. We are very sorry that we didn't clarify the simulation times.

*To reduce the systematic model errors, 91-hour simulation is conducted beginning from 0000UTC of three days ago for each day. The first 64-h of each simulation is considered as the spin-up period, the next 24-h is used for further analysis and the remaining 3-h is discarded (e.g., run one simulation from December 29, 0000 UTC (0800 BJT) to January 01, 1800 UTC (January 02, 0200 BJT), and in total 91 hours. We need the results from the January 01, 0000 BJT to 2300 BJT. From December 29, 0800 BJT to December 31, 2300 BJT is considered as the spin-up period (in total 64-h), and the results from January 02, 0000 BJT to 0200 BJT is discarded).* This has no effect on the results. Please see Lines 200-206 in the revised manuscript.

The initial and boundary conditions of chemical fields were configured using the global model output of Model for Ozone and Related Chemical Tracers (MOZART), which has been widely used (http://www/acom.ucar.edu/wrf-chem/mozart.shtml).

For the initial and boundary conditions of the model, please see Lines 183-189 in the revised manuscript.

**Question7:**

line 175 Is the 0.01 minimum value common to the original and new schemes?

**Response7:**

The minimum value is the same in the original scheme and the new scheme. Please see Lines 223-224 and Figure 2 in the revised manuscript.

**Question8:**

line 176 If one wants u = 0 at z =0 then one can set l = k(z+z0)/(1 + kz/λ). where z0 is a roughness length. There is no mention of roughness length until line 353, but it can

be an important parameter and should be different for momentum, heat and PM2.5. It is present in WRF and should be discussed. It could be linked to the 0.01 minimum in Eq (2)?

**Response8:**

We agree with you that the roughness length is an important parameter and should be different for momentum, heat, moisture and $PM_{2.5}$. Moreover, the roughness length of momentum, heat and moisture do exist in the WRF model. We did not discuss this, here, mainly consider the following reasons. 1. The experiment station is in the southern suburbs of Dezhou city (37.15°N, 116.47°E), and flat farmland around this station (Fig. R3). The underlying surface is flat, and the roughness length is considered to be zero. 2. The roughness length is in the surface layer scheme in the WRF, not in the PBL scheme. In this study, we focus on the turbulent diffusion coefficient in the PBL scheme. Therefore, we default the roughness length to zero in the calculation of mixing length, which is consistent with pervious (Blackadar, 1962; Louis, 1979; Lin et al., 2008; Pleim, 2016). We have added this information in the revised manuscript (Lines 113-116, Figure S1). The setting of the minimum value assumes that the turbulence will not disappear completely in the model.

According to your opinion, it can be said that the roughness length is considered in the setting of the minimum value. This opinion inspired me, and it may be an extremely good innovation and we need to try in the further. Because we need to observe different underlying surface to obtain different roughness length, and use these observation data to do some interesting work combined with numerical simulation.

[Figure]

**Figure R3.** Google Earthmap of the Pingyuan observational site (marked as the red pentagram). The surrounding terrain (within a range of 5 km) is shown in (b) (adapted from Ren et al., 2020).

**Question9:**

line 186 It is worth comparing the current fh and fc, with the expression one would get using MOST, if we use the standard form $1/fh = \varphi h = 1 + 5\ z/L$, and relationship with Ri from Garratt (p52), and assuming $L = 0$ for $Ri > 0.2$. A quick plot is below with fmost $= 1/\varphi h$.

[Figure]

**Response9:**

Your suggestion is very good! Actually, the fitting function of particles (i.e., $\varphi_c$) has been obtained in Ren et al (2020). It is found that the turbulent diffusion of particles is greater than that of the original turbulent diffusion in Easter China (Figure 9), which should correspond to the cases where Ri>~0.2. If we use the function you mentioned (i.e., fmost), there is no doubt that when Ri>0.2, the turbulent diffusion coefficient is equal to the minimum value (i.e., 0.01). We will get the smaller turbulent diffusion than the original scheme, which will lead to worse simulation of pollutant concentration in Easter China. The function you mentioned is similar to the cut-off function in Jia et al (2021), and when Ri exceeds a certain critical value, the turbulent diffusion is the minimum value.

**Question10:**

For moderate values (<0.2) of Ri, MOST and fh are both significantly larger than fc so that the modified TDC is significantly reduced relative to MOST or earlier assumptions with Kc = Kh.

**Response10:**

Your understanding is absolutely correct. We found that the turbulent diffusion of particles is smaller than that of heat in the original scheme when Ri<~0.2. And discuss and analyze what we have mentioned in the manuscript. Please see Section 4.3 and Figure 9 for details.

**Question11:**

For Ri > 0.25, fc > fh as noted on line 188, but how often does this occur - in reality and in the model. In the model there may be confusion between gradient and bulk Ri. Can we see a pdf of Ri values?

**Response11:**

We have added the Ri in Figure R2, and the Section 2 has been modified accordingly in the revised manuscript. Figure R2 is a time series of Ri during the observation period.

**Question12:**

line 199 "avoids the inapplicability of MOST". This "inapplicability" should be explained, maybe it is because we are not necessarily in a constant flux layer? MOST need not require fc = fh.

**Response12:**

We should describe it in more detail here, and we have reorganized the language of this part. Please see Lines 253-259 in the revised manuscript.

**Question13:**

line 208 How is "night" defined in forming these averages?

**Response13:**

We take the time when the shortwave radiation is zero as the dividing line, and night is from 18:00 on the first day to 07:00 on the second day. This part has been supplemented in the revised manuscript. Please see Lines 266-267 in the revised manuscript.

**Question14:**

line 223 Why the big difference in 2014? I am puzzled by why "relative bias" and "absolute bias" % differences are different. I assume that these are night-time values? Based on hourly data?

**Response14:**

Firstly, the pollution situation is different every year, so there must be differences in different cases, and the model cannot show a unified deviation in all cases. Secondly, we want to use a coordinate axis to explain the relative bias and absolute bias, which may be more helpful to your understanding. Figure R4 show the relative bias and absolute bias. In simple words, relative bias refers to the deviation between the original (or new) simulations and observations (i.e., the values of A or B), and the absolute bias refers to the absolute value difference between two relative biases (i.e., the value of C). And all simulated and observed pollutant concentrations are based on hourly data.

[Figure]

**Figure R4.** Schematic diagram of relative bias and absolute bias. OBS represents observation, Original and New indicate the simulation results of the original and new schemes, respectively. A and B represent the relative bias of the two schemes compared with the observation. C indicates absolute bias.

**Question15:**

From Fig 3 the relative and absolute bias values seem to be of opposite sign at some locations, A detailed definition of these quantities should be provided. I can guess but a few equations would help.

**Response15:**

The relative bias refers to the difference between the simulated value of the new scheme and observation in Fig. 4 (in the revised manuscript). The absolute bias decreases, indicating that the simulation results are improved. We believe that Figure R4 should be of great help to you.

**Question16:**

line 235 So n in Fig 4 is 31*24? I am not familiar with Taylor diagrams. Is the vertical axis from the model and the horizontal the observations?

**Response16:**

The number of points in Fig. 5 (in the revised manuscript) represents the observation stations used for comparison. The abscissa and ordinate represent the normalized standard deviation, and the arc represents the index of agreement. We can see that an "OBS" is marked on the abscissa. When the simulated statistical parameters can reach this point, indicating that the simulation is completely consistent with the observations.

**Question17:**

line 261 Can you say anything about deposition to the surface, which could be a critical removal process, or a source?

**Response17:**

The deposition process is very important for the evolution of pollutants, and we agree with you. We have supplemented the information about deposition in the model in Section 2 in the revised manuscript, please see Lines 212-219 in the revised manuscript. However, in the process of improving the model, we only changed the turbulent diffusion and did not change the dry deposition. Therefore, the change in Figure 6 is

only represents the change in pollutant concentration caused by the change in turbulent diffusion. We think it is best to use the process analysis method to quantify the contribution of each process, but this is a huge project, which is beyond the scope of this study. We can find a way to decompose each process in the model and then quantify the contribution of each process. We will do this work in the future, and we hope it can arouse your interest.

**Question18:**

line 287 There is a lot to be said about keeping models as simple as possible. What exactly do you have in mind as a " turbulence-aerosol two-way feedback module". But not needed here.

**Response18:**

Revised as suggested. Please see Lines 350-354 in the revised manuscript.

**Question19:**

line 344 This seems to be the first mention of "source" and we are expected to accept that " there is no way to use other more elaborate inventories to quantify the uncertainty caused by emissions".

**Response19:**

In Section 4, even if we cannot quantify the uncertainty caused by emission sources, we should also mention it. The emission sources process is very key to the simulation of pollutants. We are unable to make further analysis, due to the limited of the current inventory. Once we have these resources, we will further improve the current work. In fact, there are many processes affecting the evolution of pollutants, and we only make an in-depth analysis of one of the very important processes, hoping to contribute to the development of the model.

**Question20:**

line 395 Maybe tell us what the mean absolute errors in hourly or daily PM2.5 values (μg m-3) are in order to see how significant these bias improvements are.

**Response20:**

Here we express the regional average of absolute bias, which is shown in Fig. 4i-l.

**Question21:**

line 398 " Therefore, the pollutant concentration is reduced near the surface and better mixed in the whole layer, increasing the pollutant concentration in the upper level." Are there any upper level measurement to validate this effect?

**Response21:**

What we actually want to express here is that when the pollutant concentration is decreased near the surface, where does the pollutant go. We want to explain this phenomenon through models. However, we do not have vertical observations of particles for verification during all simulation periods. In recent years, we will have some vertical observation data of particles, but the latest emission source inventory has not been published, so we cannot simulate more periods. We will continue to do it in this direction and supplement our work step by step. After reviewing your professional advice, we found that there are still many ideas to be done, but not all problems can be solved at one time, we will continue to improve in the future.
* * *
Reference.

Garratt, J.R., (1992) The atmospheric boundary layer, Cambridge, UK

**References**

Beljaars, A. C. M., and Holtslag, A. A. M.: Flux parameterization over land surfaces for atmospheric models. Journal of Applied Meteorology, 30(3), 327–341. https://doi.org/10.1175/1520-0450(1991)030<0327:FPOLSF>2.0.CO;2, 1991.

Blackadar, A. K.: The vertical distribution of wind and turbulent exchange in a neutral atmosphere, J. Geophys. Res., 67, 3095–3102, https://doi.org/10.1029/JZ067i008p03095, 1962.

Chen, L., Zhu, J., Liao, H., Gao, Y., Qiu, Y., Zhang, M., Liu, Z., Li, N., and Wang, Y.: Assessing the formation and evolution mechanisms of severe haze pollution in the Beijing–Tianjin–Hebei region using process analysis, Atmos. Chem. Phys., 19, 10845-10864, https://doi.org/10.5194/acp-19-10845-2019, 2019.

Derbyshire, S. H.: Boundary-layer decoupling over cold surfaces as a physical boundary-instability. Boundary-Layer Meteorology, 90, 297–325. https://doi.org/10.1023/A:1001710014316, 1999.

Du, Q., Zhao, C., Zhang, M., Dong, X., Cheng, Y., Liu, Z., Hu, Z., Zhang, Q., Li, Y., Yuan, R., and Miao, S.: Modeling diurnal variation of surface PM2.5 concentrations over East China with WRF-Chem: impacts from boundary-layer mixing and anthropogenic emission, Atmos. Chem. Phys., 20, 2839–2863, https://doi.org/10.5194/acp-20-2839-2020, 2020.

Gao, J., Zhu, B., Xiao, H., Kang, H., Pan, C., and Wang, D., and Wang, H.: Effects of black carbon and boundary layer interaction on surface ozone in Nanjing, China, Atmos. Chem. Phys., 18, 7081–7094, https://doi.org/10.5194/acp-18-7081-2018, 2018.

Garratt, J.: The atmospheric boundary layer, Cambridge, UK, 37, 1992.

Guo, J., Miao, Y., Zhang, Y., Liu, H., Li, Z., Zhang, W., He, J., Lou, M., Yan, Y., Bian, L., and Zhai, P.: The climatology of planetary boundary layer height in China derived from radiosonde and reanalysis data, Atmos. Chem. Phys. 16, 13309–13319. https://doi.org/10.5194/acp-16-13309-2016, 2016.

Jia, W., Zhang, X., Zhang, H., and Ren, Y.: Application of turbulent diffusion term of aerosols in mesoscale model, Geophys. Res. Lett., 48, e2021GL093199, https://doi.org/10.1029/2021GL093199, 2021.

Lin, J., Youn, D., Liang, X., and Wuebbles, D. J.: Global model simulation of summertime U.S. ozone diurnal cycle and its sensitivity to PBL mixing, spatial resolution, and emissions, Atmos. Environ., 42, 8470–8483, https://doi.org/10.1016/j.atmosenv.2008.08.012, 2008.

Louis, J.: A parametric model of vertical eddy fluxes in the atmosphere, Bound.-Lay. Meteorol., 17, 187–202, https://doi.org/10.1007/BF00117978, 1979.

Louis, J., Tiedtke, M., and Geleyn, J.: A short history of the PBL parameterization at ECMWF, ECMWF, 59–79, https://www.ecmwf.int/node/10845, 1982.

Miao, Y., Liu, S., Guo, J., Huang, S., Yan, Y., and Lou, M.: Unraveling the relationships between boundary layer height and PM2.5 pollution in China based on four-year radiosonde measurements, Environ. Pollut., 243, 1186–1195, https://doi.org/10.1016/j.envpol.2018.09.070, 2018.

Pleim, J. E., Gilliam, R., Appel, W., and Ran, L.: Recent advances in modeling of the atmospheric boundary layer and land surface in the coupled WRF-CMAQ model, Air Pollution Modeling and its Application XXIV, 391–396, https://doi.org/10.1007/978-3-319-24478-5_64, 2016.

Ren, Y., Zhang, H., Wei, W., Cai, X., and Song, Y.: Determining the fluctuation of PM2.5 mass concentration and its applicability to Monin–Obukhov similarity, Sci. Total Environ., 710, 136398, https://doi.org/10.1016/j.scitotenv.2019.136398, 2020.

Savijärvi, H., and Kauhanen, J.: High resolution numerical simulations of temporal and vertical variability in the stable wintertime boreal boundary layer: a case study, Theor. Appl. Climaol., 70, 97–103, https://doi.org/10.1007/s007040170008, 2002.

Seidel, D.J., Zhang, Y., Beljaars, A., Golaz, J.C., Jacobson, A.R., and Medeiros, B.: Climatology of the planetary boundary layer over the continental United States and Europe, J. Geophys. Res. Atmos. 117, D17106. https://doi.org/10.1029/2012JD018143, 2012.

Wang, A., Li, Y., Zhao, C., Du, Q., Wang, X., and Gao, Z.: Influence of different boundary layer schemes on PM2.5 concentration simulation in Nanjing, China Environmental Sciences, 41, 2977-2992, https://doi.org/ 10.19674/j.cnki.issn1000-6923.2021.0301, 2021 (in Chinese).

Zhang, Y., Seidel, D.J., and Zhang, S.: Trends in planetary boundary layer height over Europe, J. Clim. 26, 10071e10076. https://doi.org/10.1175/JCLI-D-13-00108.1, 2013.

Zhong, J., Zhang, X., Dong, Y., Wang, Y., Liu, C., Wang, J., Zhang, Y., and Che, H.: Feedback effects of boundary-layer meteorological factors on cumulative explosive growth of PM2.5 during winter heavy pollution episodes in Beijing from 2013 to 2016, Atmospheric Chemistry and Physics, 18, 247–258. https://doi.org/10.5194/acp-18-247-2018, 2018.

Zhong, J., Zhang, X., Wang, Y., Wang, J., Shen, X., Zhang, H., Wang, T., Xie, Z., Liu, C., Zhang, H., Zhao, T., Sun, J., Fan, S., Gao, Z., Liu, Y., Wang, L.: The two-way feedback mechanism between unfavorable meteorological conditions and cumulative aerosol pollution in various haze regions of China. Atmospheric Chemistry and Physics, 19, 3287–3306. https://doi.org/10.5194/acp-19-3287-2019, 2019.

**Response to Referee #2**

**General Comments:**

I fell that my major concerns are appropriately addressed in the revised version. But I think the manuscript can be further improved. So my recommendation is publication in ACP after minor revisions.

**Response:**

Thank you again for your positive comments and valuable suggestions to improve the quality of our manuscript. Based on these comments and suggestions, we have made careful modifications to our pervious manuscript, and the detailed point-by-point responses are listed below.

**Specific Comments:**

1) For the value of TDC, the authors state in line 186-188 that "When Ri is greater than ~0.2, the TDC of particles is greater than that of heat, which may reduce pollutant concentration". This statement actually tell us that the TDC of particles is greater than that of heat when Ri is greater than ~0.2. However, the TDC of particles is smaller than that of heat when Ri is smaller than ~0.2. Therefore the result that the new scheme of TDC for particles can reduce the overestimated PM2.5 concentration implies that Ri is greater than ~0.2 in the most part of nighttime. So I suggest that the authors should add a paragraph to discuss this issue (it's better to give the information about the statistics of the value of Ri), which will help the readers to understand the results more easily.

**Response:**

We agree with your understanding. According to the fitting results of our previous observation data, the TDC of particles is greater than that of heat in the original scheme when Ri is greater than ~0.2. Although the Ri is not an input/output quantity in the model, the change in TDC is considered to be caused by the modification of $f(Ri)$ when other physical quantities remain unchanged.

According to your suggestion, we have added discussion in the revised manuscript (Lines 390-393 and Lines 407-409).

Once again, we have substantially revised the content of data and methods (i.e., including information about Ri), and provide a flow chart (Figure 2), so that readers can understand our results more clearly. Please see Section 2 in the revised manuscript.

**Technical Comments:**

1) Line 287: What is "turbulence barrier effect"? Please give the explanation.

**Response:**

Revised as suggested, we have added this concept to the revised draft (Lines 59-61).

"It means turbulence may disappear at certain heights forming a laminar flow as if there is a barrier layer hindering the transmission up and down during the heavy pollution episodes". This phenomenon as the turbulent barrier effect, for detailed discussion, please refer to Ren et al. (2021).

2) Line 288: What is "HPEs"? Please give the specification.

**Response:**

Revised as suggested (Lines 354).

HPEs refers to "Heavy Pollution episodes".

3) Fig. 5: Does each panel show the result for one day, or monthly mean? Does each panel show the result for multi-point mean, or area mean? Please add the specifications in the figure caption.

**Response:**

Revised as suggested (Fig. 6).

Figure 6 shows the situation in Hefei in the previous manuscript (Fig. R1). To be consistent with the statement of PBL height, we still selected Anqing (Fig. R2) and Fuyang (Fig. R3) stations. In addition, we selected the results of three typical cities (i.e.,

Hefei, Nanjing (Fig. R4) and Shanghai (Fig. R5)) in Eastern China as auxiliary verification, and added these results to the supplementary materials.

[Figure]

**Figure R1.** Time-height cross sections for the difference of $PM_{2.5}$ concentration between original and new schemes (i.e., the new scheme minus the original scheme) within the PBL in Hefei from 2013 to 2017. The gray line indicates the PBLH.

[Figure]

**Figure R2.** Similar to Figure R1, but in Anqing.

[Figure]

**Figure R3.** Similar to Figure R1, but in Fuyang.

[Figure]

**Figure R4.** Similar to Figure R1, but in Nanjing.

[Figure]

**Figure R5.** Similar to Figure R1, but in Shanghai.

**References**

Ren, Y., Zhang, H., Zhang, X., Wei, W., Li, Q., Wu, B., Cai, X., Song, Y., Kang, L., and Zhu, T.: Turbulence barrier effect during heavy haze pollution events, Sci. Total Environ., 753, 142286, https://doi.org/10.1016/j.scitotenv.2020.142286, 2021.

---

## Author Response (AR3)

**Response to Editor**

The authors sincerely thank the reviewers' professional evaluation and valuable suggestions. According to suggestions from Editor and reviewers, we have made corresponding corrections to our previous manuscript, and the detailed point-by-point responses are listed below

**Response to Referee #1**

**RE:** Impact of modified turbulent diffusion of PM2.5 aerosol in WRF-Chem simulations in Eastern China

**Author(s):** Wenxing Jia and Xiaoye Zhang

**MS No.:** acp-2021-435: MS type: Research article; Iteration: Revised submission

**General Comments** - Peter Taylor

**Question1:**

The authors have made a number of changes and have provided detailed responses to questions raised about earlier versions. I still have concerns about the modified stability function used, Eq 4, in connection with the Turbulent Diffusion Coefficient (TDC) and feel that the degree of improved performance is exaggerated. Despite Response 14, I am not really sure whether the % improvements are C/A or C/OBS values. Providing the bias values of both the original and the new schemes might be clearer. From Fig 4, Relative Bias is given as (NEW-OBS)/OBS and it is clear that individual stations have a wide range of bias % and that modest improvements of order 10% (line 453) in the mean over all stations may not warrant the claim, line 286 that "the new scheme can significantly reduce the degree of overestimation", and similar on line291. Also line 452 could make it clear whether these are % reductions of % overestimates rather reduction in the % overestimates themselves? Reduced by, or reduced to?

**Response1:**

Thank you again for your professional comments and valuable suggestions to improve the quality of our manuscript. Based on these comments and suggestions, we have made careful modifications to our pervious manuscript, and the detailed point-by-point responses are listed below.

Indeed, we agree with you that the coefficients of the function can continue to be corrected in the future with the support of more observation data. However, the role of turbulent diffusion is very important. Other processes may affect the simulation of pollutants, such as dry deposition process and emissions. Much more researches need to be done in this field in the future. Percentage improvement here refers to the improvement of the new scheme compared with the original scheme. Actually, the absolute bias represents the difference between the new scheme and original scheme. According to your suggestion, we have modified this part. The percentage (%) in Line 452 refers to the reduction in the overestimates themselves, and it is reduced by.

**Question2:**

While I appreciate the need to reference relevant prior work I am surprise that a paper on a relatively narrow topic needs about 60 references. The paper is well written but will need some language editing.

**Response2:**

Thank you very much for your affirmation. In the description part of the model, there are many parameterization schemes, and each scheme needs corresponding literature, which may lead to too many literatures. According to your suggestions, we have modified the language and deleted some unimportant references.

**Minor points**

**Question1:**

p5 Line 109 It may be misleading to say that the roughness is considered as zero. As the authors note in Response 8 to questions on the previous version, WRF does treat the PBL and surface layer and, unfortunately, the PBL code ignores z0. Although no

changes have been made to the surface layer code it probably does involve a roughness length based on land use maps. WRF boundary layer modules, MYNN and YSU make use of z0 values based on land use. At this stage just avoid discussing z0 unless you plan to dig into the WRF code and find out.

**Response1:**

Thanks so much for all your helpful advice and info! We have revised the statements in this part.

**Question2:**

p5 line 120 Does this suggest that Ri values in the original field data were based on (60m - 10m) differences? Do these really give representative "gradients"

**Response2:**

The disturbance of the early time of the GPS sounding balloons taking off would cause uncertainty of the mass concentration of $PM_{2.5}$ near the ground, so we selected 10 m as the lower height to avoid this. According to the constant flux layer hypothesis, the upper level should be within the surface layer. For convenience of calculation, we rounded the height difference to 50 m. Therefore, 60 m was selected to be the higher level for calculating the vertical gradient of $PM_{2.5}$ concentration.

**Question3:**

p6 line 141 I did not have time to look into the "L-band radiosonde system" but assume it transmits 1Hz data as it rises. The issue is what vertical z resolution does this represent - what is the rise rate?

**Response3:**

The resolution of L-band radiosonde data is 1 Hz, and the rise rate of data in this study is about 6m $s^{-1}$. If necessary, we can provide some raw data. Thanks.

**Question4:**

p7 line 185 Not clear what the 8 months are, 4 with the original model and 4 with the new?

**Response4:**

Yes, your understanding is correct.

**Question5:**

p7 line 185 Re Fig 2, I am just curious whether kstart = 1 always in these runs?

**Response5:**

Yes, $k_{start}$ is always the same in all runs, and $k_{start} = 1$.

**Question6:**

p8 line 210 ++ It is fine to use fine grid finite differences to approximate gradient Ri. Formally a bulk RiB would be based on two widely separated levels, one of which is normally the surface, and should be specific to those levels. As Garratt (1992, p37) notes, politely, some authors use the bulk term incorrectly.

**Response6:**

You're quite right. In practice, the gradient Richardson number is often approximated in finite difference form and the resulting parameter is sometimes referred to as the bulk Richardson number. In our model setting, we have encrypted the number of vertical layers below 2 km, which can better analyze the structural characteristics of the boundary layer. Please see Lines 163-165 in the revised manuscript.

**Question7:**

p11 line 269 See general comment. Need to make clear what "mean absolute bias" means. Is it the mean of an absolute value <|OBS-Model|> or just <OBS-Model>, where <...> means "mean".

**Response7:**

We are very sorry for the confusion caused by the unclear expression. The "mean absolute bias" is the mean value of absolute bias. We have modified the corresponding statements. The absolute bias here refers to the deviation between the new scheme and old scheme relative to the observation. And the calculation formula of absolute bias is $AB = \left| RB_{new} \right| - \left| RB_{original} \right|$, where $\left| RB_{new} \right|$ and $\left| RB_{original} \right|$ represent the relative bias of new and original schemes, respectively. The calculation formula of relative bias is $RB = \left( \overline{X_{sim}} - \overline{X_{obs}} \right) \Big/ \overline{X_{obs}} \times 100\%$ .

**Question8:**

p15 line 393 Are model values of dry deposition available? Do they play a significant role in the PM2.5 budget?

**Response8:**

In the current model, we really do not separate the contribution of each process. However, based on the previous research results of individual cases using process analysis (Chen et al., 2019), it can be seen that the process of dry deposition has a certain impact, but the contribution of dry deposition is less than those of emission and turbulent diffusion. Next, we hope to analyze the long-term heavy pollution episodes in detail through the method of process analysis.
* * *
**References**

Chen, L., Zhu, J., Liao, H., Gao, Y., Qiu, Y., Zhang, M., Liu, Z., Li, N., and Wang, Y.: Assessing the formation and evolution mechanisms of severe haze pollution in the Beijing–Tianjin–Hebei region using process analysis, Atmos. Chem. Phys., 19, 10845-10864, https://doi.org/10.5194/acp-19-10845-2019, 2019.